# Transcriptional pattern enriched for synaptic signaling is associated with shorter survival of patients with high-grade serous ovarian cancer

Arkajyoti Bhattacharya[1†], Thijs S Stutvoet[1†], Mirela Perla[1†], Stefan Loipfinger[1], Mathilde Jalving[1], Anna KL Reyners[1], Paola D Vermeer[2], Ronny Drapkin[3], Marco de Bruyn[4], Elisabeth GE de Vries[1], Steven de Jong[1], Rudolf SN Fehrmann[1*]

[1]Department of Medical Oncology, University Medical Center Groningen, University of Groningen, Groningen, Netherlands; [2]Cancer Biology and Immunotherapies Group, Sanford Research, Sioux Falls, United States; [3]Penn Ovarian Cancer Research Center and Basser Center for BRCA, University of Pennsylvania, Perelman School of Medicine, Philadelphia, United States; [4]Department of Obstetrics and Gynecology, University Medical Center Groningen, University of Groningen, Groningen, Netherlands

*For correspondence:
r.s.n.fehrmann@umcg.nl

†These authors contributed equally to this work

Competing interest: The authors declare that no competing interests exist.

## eLife Assessment

This **valuable** study uses consensus-independent component analysis to highlight transcriptional components (TC) in high-grade serous ovarian cancers (HGSOC). The study presents a **convincing** preliminary finding by identifying a TC linked to synaptic signaling that is associated with shorter overall survival in HGSOC patients, highlighting the potential role of neuronal interactions in the tumour microenvironment. This finding is corroborated by comparing spatially resolved transcriptomics in a small-scale study; a weakness is it being descriptive, non-mechanistic, and requires experimental validation.

**Abstract** Bulk transcriptomic analyses of high-grade serous ovarian cancer (HGSOC) so far have not uncovered potential drug targets, possibly because subtle, disease-relevant transcriptional patterns are overshadowed by dominant, non-relevant ones. Our aim was to uncover disease-outcome-related patterns in HGSOC transcriptomes that may reveal novel drug targets. Using consensus-independent component analysis, we dissected 678 HGSOC transcriptomes of systemic therapy naïve patients—sourced from public repositories—into statistically independent transcriptional components (TCs). To enhance c-ICA's robustness, we added 447 transcriptomes from non-serous histotypes, low-grade serous, and non-cancerous ovarian tissues. Cox regression and survival tree analysis were performed to determine the association between TC activity and overall survival (OS). Finally, we determined the activity of the OS-associated TCs in 11 publicly available spatially resolved ovarian cancer transcriptomes. We identified 374 TCs, capturing prominent and subtle transcriptional patterns linked to specific biological processes. Six TCs, age, and tumor stage stratified patients with HGSOC receiving platinum-based chemotherapy into ten distinct OS groups. Three TCs were linked to copy-number alterations affecting expression levels of genes involved in replication, apoptosis, proliferation, immune activity, and replication stress. Notably, the TC identifying patients with the shortest OS captured a novel transcriptional pattern linked to synaptic signaling, which was active in tumor regions within all spatially resolved transcriptomes. The association

between a synaptic signaling-related TC and OS supports the emerging role of neurons and their axons as cancer hallmark-inducing constituents of the tumor microenvironment. These constituents might offer a novel drug target for patients with HGSOC.

## Introduction

Epithelial ovarian cancer encompasses five primary histological subtypes, with HGSOC constituting about 75% of all cases (*Lheureux et al., 2019*). The standard treatment for HGSOC diagnosed at stage IIB and beyond involves a combination of surgery and chemotherapy, primarily using platinum-based compounds and taxanes (*NCCN Guidelines, 2019*; *Wright et al., 2016*). While initial chemotherapy results in tumor response in most patients with HGSOC, there is a very high recurrence rate (*Corrado et al., 2017*). The addition of poly-ADP ribose polymerase and vascular endothelial growth factor A inhibitors to chemotherapy for subsets of patients currently results in a 5 y disease-specific overall survival (OS) rate of approximately 45% for patients with HGSOC. This rate has hardly improved in the last three decades (*Ledermann, 2016*; *Wang et al., 2018*; *Wu et al., 2019*; *Tewari et al., 2019*). Therefore, new insights into the complex biology underlying HGSOC are urgently needed to develop more effective treatment strategies.

Previous studies using bulk transcriptomes of patients with HGSOC have identified expression-based molecular subtypes. However, these subtypes did not provide insights that have translated into novel drug targets (*Bell et al., 2011*; *Tothill et al., 2008*; *Verhaak et al., 2013*). A common limitation of such studies is their reliance on bulk transcriptomes, containing both tumor cells and tumor microenvironment (TME) components, thus reflecting the average transcriptional patterns of the combination of all biological processes present in the tumors. This averaging often masks subtle transcriptional patterns pivotal to understanding HGSOC biology, especially when these are overshadowed by dominant patterns from other less relevant (non-)biological processes (*Chen et al., 2011*). Consensus-independent component analysis (c-ICA) offers an alternative by decomposing such bulk transcriptomes into statistically independent transcriptional patterns (i.e. transcriptional components; TCs) (*Kong et al., 2008*; *Chiappetta et al., 2004*). This approach reveals both dominant and subtle patterns and provides a measure of TC activity for each sample (*Biton et al., 2014*).

In the present study, our aim was to utilize c-ICA to dissect HGSOC transcriptomes to identify as many TCs associated with patient OS as possible, which could reveal potential novel drug targets.

## Methods

See the appendix for the extended methods.

### Data acquisition

Raw microarray bulk transcriptomes and clinicopathological details for patients with HGSOC, low-grade serous ovarian cancer (LGSOC), non-serous ovarian cancer, and benign ovarian tissues were sourced from the Gene Expression Omnibus (GEO)(*Clough and Barrett, 2016*). We exclusively utilized transcriptomes generated from primary tumor samples. Our analysis was confined to samples on the Affymetrix HG-U133 Plus 2.0 platform (GEO accession identifier: GPL570) and excluded cell line samples. The datasets were pre-processed and quality controlled as previously described (*Fehrmann et al., 2015*). Furthermore, for comprehensive analyses, we incorporated transcriptomes from five distinct resources: the Cancer Cell Line Encyclopedia (CCLE, n=969), Genomics of Drug Sensitivity in Cancer (GDSC, n=959), Gene Expression Omnibus (GEO, n=13,810), and The Cancer Genome Atlas (TCGA, n=8150), and spatially resolved transcriptomes from 10xGenomics (*Clough and Barrett, 2016*; *Barretina et al., 2012*; *Yang et al., 2013*; *Barrett et al., 2013*).

### Consensus-independent component analysis (c-ICA)

To preprocess the bulk transcriptome data, we applied a whitening transformation to prepare it for subsequent analysis. Consensus-ICA was conducted as described previously (*Knapen et al., 2024*). The output of a c-ICA includes two matrices: (i) transcriptional components (TCs) with gene weights, where each weight within the TC represents both the direction and magnitude of its effect on the

expression levels of each gene, and (ii) a consensus mixing matrix (MM) with its coefficients representing the activity scores of TCs across samples.

## Survival analysis

To discern the relationship between TC activity and patient OS, a univariate Cox proportional hazards analysis was conducted on a select group of patients with available follow-up data (n=541, *Supplementary file 1*). In addition, a multivariate Cox proportional hazards analysis was carried out, including covariates such as age, stage, debulking status, and tumor grade. This latter analysis was based on a subset of patients with comprehensive clinicopathological data available (n=373, *Supplementary file 1*). We implemented a multivariate permutation framework encompassing 10,000 permutations to mitigate the risk of false discoveries. We established the acceptable false discovery rate (FDR) at 1%, maintaining an 80% confidence level, applicable for both the univariate and multivariate analyses.

## Survival tree analysis

We performed a survival tree analysis to delineate groups of patients with HGSOC treated with platinum-based chemotherapy based on distinct transcriptional and clinicopathological attributes. The analysis utilized activities of TCs associated with OS (either from univariate or multivariate survival analysis as mentioned in supplementary methods) in conjunction with relevant clinicopathological factors, such as age, tumor stage, debulking status, and grade, as potential classifiers. We divided patients into two subsets using every plausible cut-off point for each classifier and compared the resulting survival curves employing the log-rank statistic. Consequently, the division was based on the most significant classifier at its optimal cut-off based on the smallest p-value of the log-rank test mentioned above. This divisional process was successfully reiterated on the derived subsets until any of the following stipulated conditions was satisfied: (*i*) the total patient count across both subsets fell below 50, (*ii*) the collective number of uncensored events in both subsets was <25, or (*iii*) one of the subsets contained <17 patients. To gauge the stability of our classifiers, we performed 20,000 iterations, randomly selecting 80% of the patient group in each iteration. The significance-based ranks of classifiers in these iterations were correlated with those from the primary survival tree.

## Associating the identified transcriptional components with biological processes

To discern the biological processes associated with the TCs, we adopted a multifaceted approach encompassing (*i*) Transcriptional Adaptation to Copy Number Alterations (TACNA) profiling, targeting the identification of TCs that reflect the downstream implications of copy number alterations (CNAs) on gene expression levels (*Bhattacharya et al., 2020*); (*ii*) Execution of gene set enrichment analysis (GSEA) for each TC, utilizing gene set collections (n=16) from The Human Phenotype Ontology (The Monarch Initiative), the Mammalian Phenotypes (Mouse Genome Database), and the Molecular Signatures Database (MsigDB) *Subramanian et al., 2005*; *Köhler et al., 2019*; (*iii*) The formation of co-functionality networks on the top and bottom genes of each TC, achieved using the GenetICA methodology, accessible via https://www.genetica-network.com (*Urzúa-Traslaviña et al., 2021*). For clusters comprising ≥5 genes, the enrichment of the predicted functionality was quantified. This served as the foundation for determining the biological process associated with the TC being examined.

## Cross-study transcriptional component projection

To determine whether a biological process captured by an identified TC is also active in other cancer types and to investigate if it is more active in tumor cells or in the TME, we collected raw expression profiles from multiple sources: the CCLE, n=969, GDSC, n=959, GEO, n=13,810, and TCGA, n=8150 (*Clough and Barrett, 2016*; *Barretina et al., 2012*; *Yang et al., 2013*; *Barrett et al., 2013*). While the CCLE and GDSC datasets comprise cell line profiles across many solid and hematologic malignancies, the GEO and TCGA datasets offer an extensive set of bulk transcriptomes derived from patient samples spanning 27 tumor types. We pre-processed the raw data as previously described (*Bhattacharya et al., 2020*). Next, we projected the TCs identified via c-ICA onto the cell line expression profiles from CCLE and GDSC and the patient-derived expression profiles from GEO and TCGA. This projection methodology has been described in more detail previously (*Bhattacharya et al., 2020*). To identify potential variations in the activity scores of the TCs, we compared the activity scores among

cell lines and samples derived from patients within all four repositories. We used an absolute activity score threshold of 0.05 for each TC to pinpoint outlier cell lines and patient tumors with heightened activity.

## Determination of spatial transcriptomic profiles' significant activity locations for individual transcriptional components

To further assess whether a biological process captured by an identified TC is more active in tumor cells or in the TME, we collected publicly available spatial resolved transcriptomic profiles of ovarian cancer samples. Eight were sourced from GEO (study ID GSE211956), and three were sourced from the public dataset repository of 10xGenomics (see supplementary methods for details) generated using the 10xGenomics Visium platform. The samples were from patients with HGSOC, serous papillary, and endometrioid ovarian cancer. Activity for each TC across every location within the spatial samples was ascertained through the cross-study projection methodology referred to in the previous method section (*Bhattacharya et al., 2020*). We incorporated a permutation-driven approach to discern the markedly active areas within the spatial samples for each TC. We derived a null distribution of activities for each TC-location pairing by performing 3000 permutations and subsequent projections. The p-value of each observed TC activity quantifies the significance of the deviation of the TC activity at a given location from its baseline null distribution. After this, we visualized the z-transformed p-values using a heatmap, followed by obtaining colocalization scores for each combination of TCs in the spatial transcriptomic profiles for each ovarian cancer sample (*Grisanti Canozo et al., 2022*). This visualization aided in highlighting the areas with notable activity aligned against the stained representation of the tissue sample.

## Results

### An integrated data set containing 1125 bulk transcriptomes from ovarian tissues

We curated 1193 bulk transcriptomes from the GEO, including patients with HGSOC, LGSOC, non-serous ovarian cancer, and benign ovarian tissues (*Clough and Barrett, 2016*). These were extracted from 32 distinct studies and represented the entire spectrum of ovarian cancer types, stages, and grades, and included 43 samples from non-malignant ovarian tissue. Pre-processing, which included removing duplicates and quality checks, culminated in a refined dataset of 1125 samples (*Fehrmann et al., 2015*). *Supplementary file 1*; *Supplementary file 2* provide detailed breakdowns of these samples, showcasing the comprehensive coverage of ovarian cancer types, stages, and grades within this dataset. The ovarian cancer dataset comprised bulk transcriptomes of patients with HGSOC

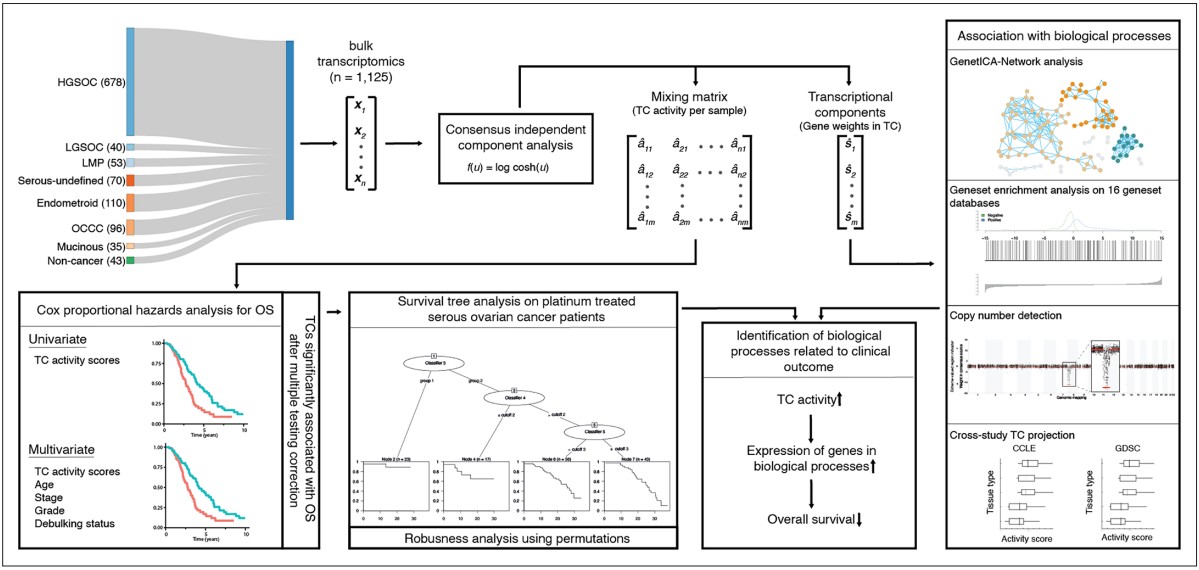

**Figure 1.** Workflow indicates the data acquisition and relations between the methods.

(n=678), other serous (n=110), endometrioid (n=110), and clear-cell ovarian cancer samples (n=96). Additionally, for 541 patients, comprehensive survival data was available, as well as additional clinicopathologic information, including age, grade, stage, subtype, treatment schedule, and debulking status for 373 patients (*Figure 1*).

## Consensus-independent component analysis identifies 374 transcriptional components (TCs)

c-ICA on the 1125 bulk transcriptomes revealed 374 independent TCs. Notably, 135 TCs captured the impact of copy number alterations on gene expression levels. Each TC displayed enrichment for at least one gene set from the 16 gene set collections, with an absolute Z-score of more than two. For example, the number of enriched gene sets from the Hallmark gene set collection in an individual TC ranged from zero to 28 enriched gene sets (interquartile range 3–7). The median top Z score for Hallmark gene sets was 3.21 (range 1.55–37.54, interquartile range 2.6–4.25). A comprehensive database, including all TCs and GSEA outcomes, has been made accessible at http://transcriptional-landscape-ovarian.opendatainscience.net.

The activities of 13 TCs were associated with patient overall survival (OS) in a univariate analysis, with an additional TC (TC166) identified in a multivariate analysis accounting for age, stage, debulking, and tumor grade. Combined, these 14 OS-associated TCs were enriched for gene sets

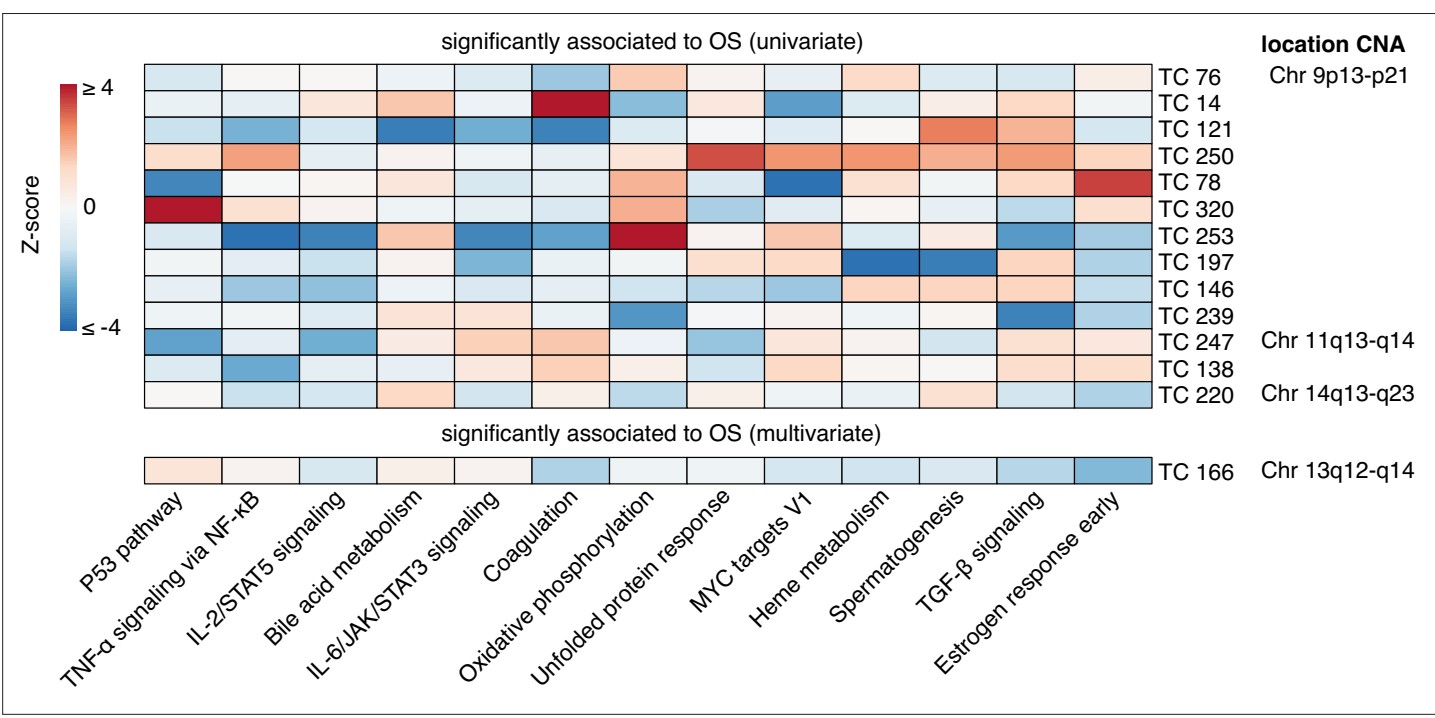

**Figure 2.** Enrichment heatmap of hallmark gene sets in transcriptional components associated with patient overall survival. Gene Set Enrichment Analysis for 14 transcriptional components (TCs) associated with overall survival (OS) identified through univariate or multivariate survival analyses are presented. Only Hallmark gene sets with significant enrichment (Bonferroni-corrected p-value) for at least one TC are shown. The heatmap displays Z-scores, which indicate the relative enrichment strength, with values truncated at a maximum of 4 for visualization purposes. The gene sets were clustered based on Pearson correlation using the Ward D2 method, providing insights into related biological processes captured by different TCs. In the right column, chromosomal locations of copy number alterations (CNAs) are shown, reflecting the downstream effects on gene expression that each TC captures. This integration of CNA information highlights the biological relevance of each TC and its contribution to gene expression variability and patient outcomes.

The online version of this article includes the following figure supplement(s) for figure 2:

**Figure supplement 1.** Association of OS-related TCs with clinicopathologic parameters.

**Figure supplement 2.** Enrichment heatmap for the KEGG gene set collection in OS-related TCs.

**Figure supplement 3.** Enrichment heatmap for the REACTOME gene set collection in overall survival (OS)-related TCs.

associated with diverse biological processes and clinicopathological characteristics, with four TCs capturing the effects of copy number alterations on gene expression levels.

## The activities of six transcriptional components are associated with patient overall survival

For a selected subset of 541 patients—including HGSOC, LGSOC, and non-serous ovarian cancer—with available OS information (*Supplementary file 1*), 13 TC activities displayed an association with OS univariately (false discovery rate of 5%, confidence level of 80% in permutation-based multiple testing framework *Supplementary file 3*; *Figure 2*). For patients with serous ovarian cancer, treated with platinum-based therapy (n=301, *Supplementary file 1*), lower activity of one additional TC (TC166) was associated with worse OS independent of age, stage, debulking, and tumor grade (*Supplementary file 4*). Combined, these 14 OS-associated TCs were enriched for gene sets associated with diverse biological processes and clinicopathological characteristics. Four of these TCs captured the downstream effects of CNAs on gene expression levels (*Figure 2*, *Figure 2—figure supplements 1–3*). Survival tree analysis identified ten groups of patients with platinum-treated HGSOC based on the activity of six OS-associated TCs and the presence of two clinicopathological characteristics, namely age and stage (*Figure 3*, *Supplementary file 5*, median robustness statistic of survival tree = 0.52, interquartile range = 0.36–0.69). The survival tree demonstrated good classification power (concordance statistic = 0.72, standard error = 0.021). As expected, patients were divided into separate survival groups based on stage (1/2 vs 3/4) and age (<53.7 vs ≥53.7 y). The most significant difference in OS was observed between the cohorts with low and high TC121 activity (*Supplementary file 5*). Patients with high TC121 tumor activity exhibited the shortest OS, also observed for the subset of patients with advanced-stage HGSOC (*Figure 3—figure supplement 1*, *Supplementary file 6*). *Figure 3—figure supplement 2* indicates that TC121 activity is highest in patients with HGSOC compared to other ovarian cancer subtypes. Notably, all subtypes contain subsets of samples with elevated TC121 activity. These robust associations with OS for TC121 in these two subsets of patients indicate the relevance of TC121, irrespective of stage.

## Distinct biological processes show enrichment in the transcriptional components associated with overall survival

Three of the six TCs associated with OS—TC166, TC247, and TC76—captured the effects of CNAs on the expression levels of genes mapping to chromosome regions 13q12-q14, 11q13-q14, and 9p13-p21, respectively (*Figure 3—figure supplement 3*, *Supplementary file 7*; *Wang et al., 2018*). The higher activity of TC166 was associated with better OS, whereas the higher activities of TC121, TC247, TC250, TC76, and TC146, were associated with worse OS. Among the 14 OS-associated TCs, only TC166 showed a significant association with OS in an independent cohort of patients with ovarian clear cell carcinoma (Bonferroni corrected p-value <0.05; see supplementary methods and *Supplementary file 8*: *Bolton et al., 2022*). The top genes from TC166 were enriched for genes involved in replication and apoptosis. The chromosomal region 13q12-q14 linked to the TC166 contains the tumor suppressor genes retinoblastoma 1 (*RB1*) and Breast Cancer Type 2 Susceptibility Protein (*BRCA2*). Loss of heterozygosity of this chromosomal region is frequently observed in both sporadic and hereditary serous ovarian cancers (*Huang et al., 2012*; *Jongsma et al., 2002*). The top genes from TC247 were enriched for genes involved in proliferation and immune cell activation, TC76 in replication stress, TC250 in extracellular matrix (ECM) interactions, and TC146 in neurotransmitter signaling.

Intriguingly, the top 100 genes in TC121 revealed a co-functional cluster enriched for genes involved in synaptic signaling, with the corresponding proteins reported to localize to the synaptic membrane of neurons (*Figure 4*). Among these were pre-synaptic protein neurexin-1 (*NRXN1*) and its post-synaptic ligand leucine-rich repeat transmembrane protein 2 (*LRRTM2*), which regulates excitatory synapse formation (top 20 genes are described in *Supplementary file 9*, for more details: http://transcriptional-landscape-ovarian.opendatainscience.net) (*Ko et al., 2009*; *de Wit et al., 2009*). Furthermore, this co-functional cluster included neuron-specific synaptic structure proteins, neurofilament light, and medium chain. Moreover, genes encoding for potassium ion channel proteins integral to membrane repolarization during synapse signal transduction carried high weights in TC121. These genes included *KCNC1*, *KCNN2*, and *KCNIP1* (*Bourdeau et al., 2011*; *Ried et al., 1993*; *Willis et al.,*

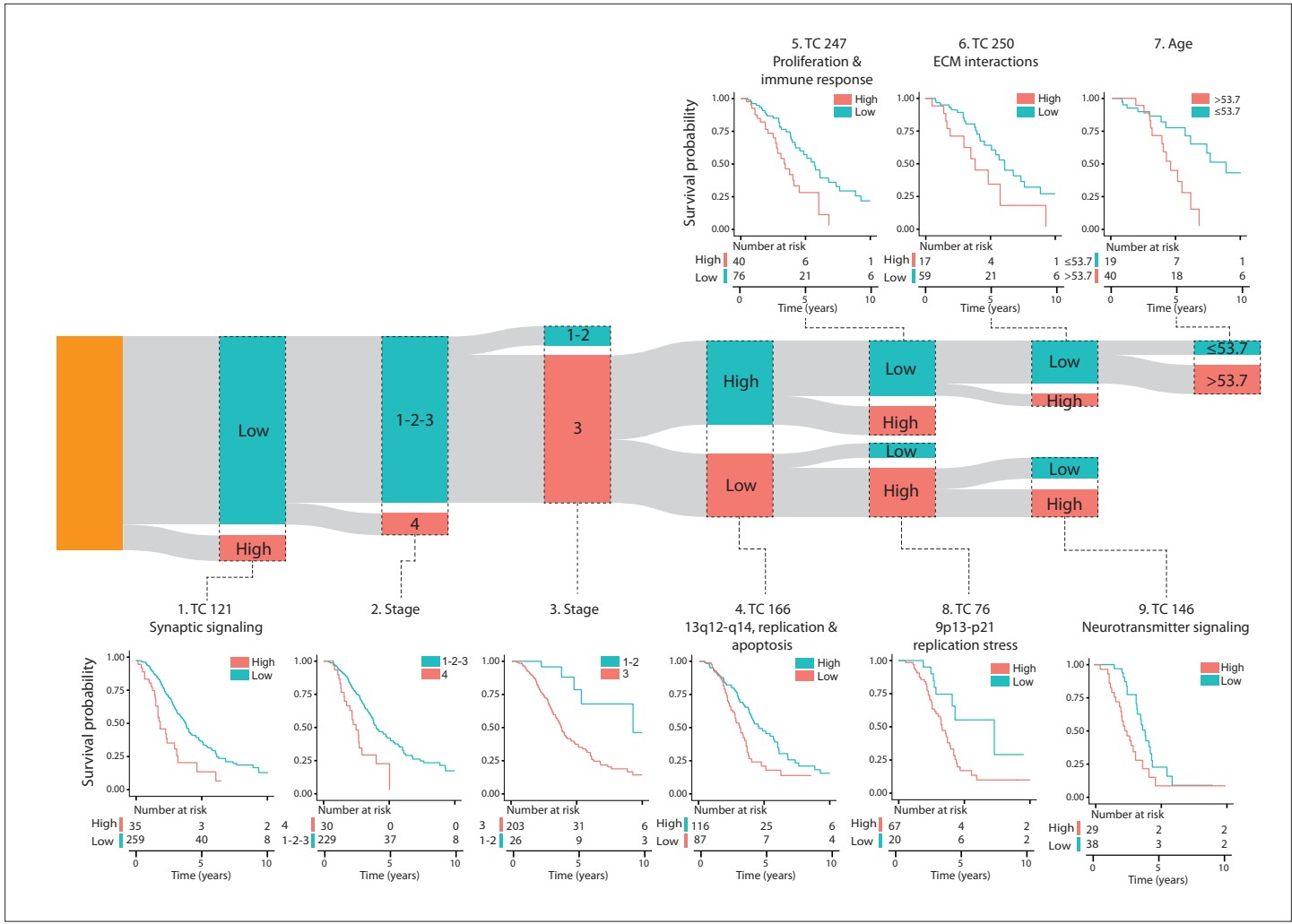

**Figure 3.** Survival tree analysis of patients with platinum-treated HGSOC defines survival cohorts with distinct clinicopathologic and biological characteristics. The results of the survival tree analysis of 294 patients with high-grade serous ovarian cancer (HGSOC) treated with platinum-based chemotherapy are presented. The analysis utilized 14 transcriptional components (TCs) associated with overall survival (OS), along with other clinicopathologic factors, including age, tumor stage, grade, and debulking status. The resulting tree identified nine distinct survival cohorts, each represented as a bar in the Sankey diagram, where the bar height corresponds to the number of patients in each cohort. Kaplan-Meier survival curves with accompanying number-at-risk tables are shown for each cohort, with survival data censored at 10 y. The names of the survival cohorts were based on enriched biological processes in the TCs, as determined by the chromosomal location of genes captured by a TC, GSEA, and co-functionality analysis of the top genes. The p-values in the Kaplan-Meier plots were derived from log-rank tests comparing survival distributions between groups. Abbreviations: TC = transcriptional component, ECM = extracellular matrix.

The online version of this article includes the following figure supplement(s) for figure 3:

**Figure supplement 1.** Survival tree analysis of patients with advanced-stage, high-grade serous ovarian cancer (HGSOC) defines survival cohorts with distinct clinicopathologic and biological characteristics.

**Figure supplement 2.** The activity of TC121 in bulk transcriptomes of patients with different subtypes of ovarian cancer.

**Figure supplement 3.** Three overall survival (OS)-associated TCs capture the transcriptional effect of copy number alterations.

*2017*). Several genes in TC121 encoded proteins related to glutamate receptor signaling, including *GRIN2C* and *SLC7A10* (*Ehmsen et al., 2016*). In line with this proposed function, high activity of TC121 was observed in neuroblastoma cell lines but not in ovarian or central nervous system cancer cell lines in the GDSC and CCLE datasets (*Figure 5A*, *Figure 5—figure supplements 1 and 2*). In the GEO and TCGA datasets, high activity of TC121 was observed in glioblastoma multiforme and lower-grade glioma but not in ovarian cancer patient samples (*Figure 5B*).

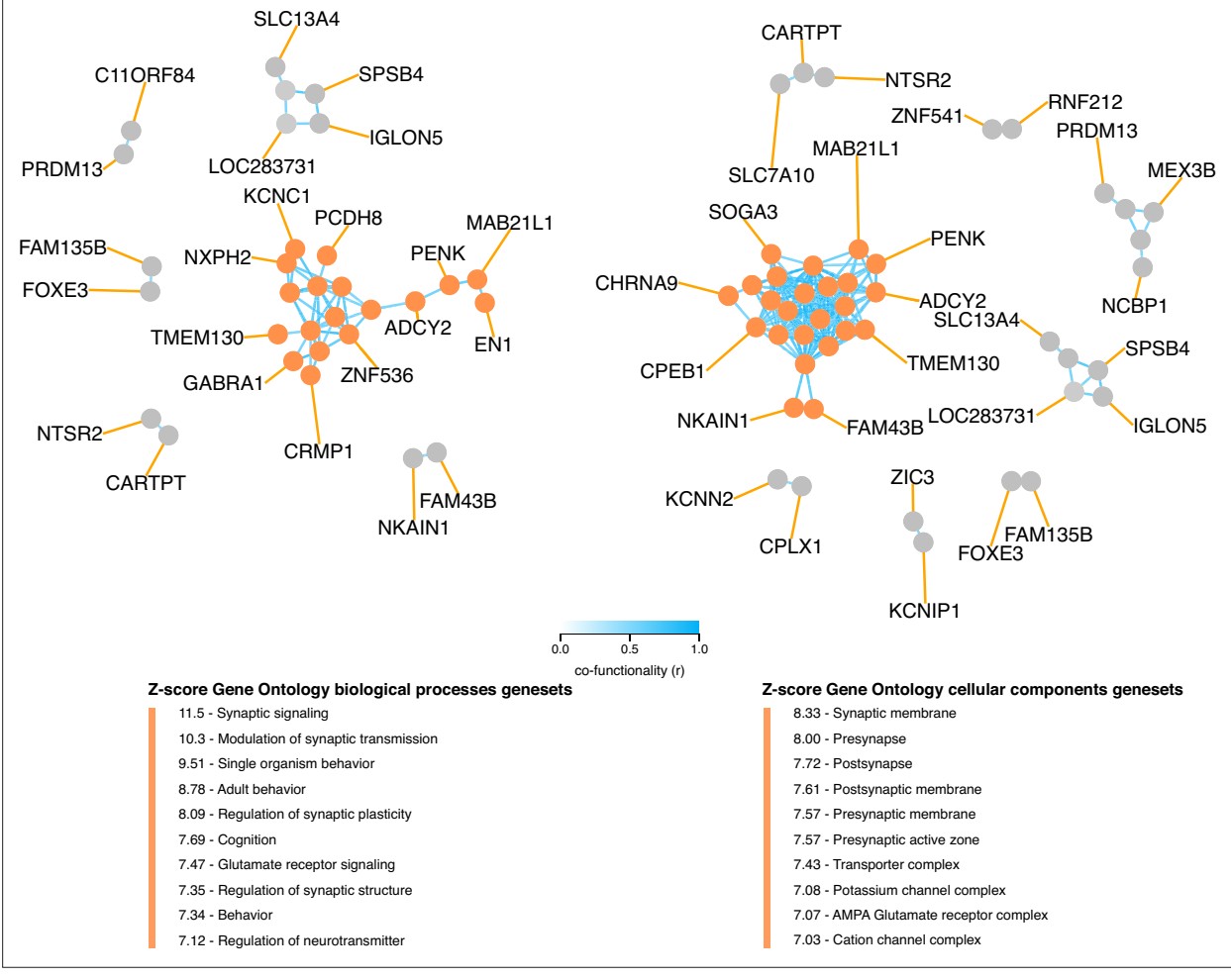

**Figure 4.** Co-functionality network of top 100 absolute weighted genes in TC121. Co-functionality network for the top 100 genes with the highest absolute weights in TC121 is presented. Genes were clustered based on predicted co-functionality (*r*>0.7) across datasets, with clusters identified using both Gene Ontology (GO) Biological Processes and Cellular Components databases. One primary cluster, containing more than five genes, exhibited strong enrichment for synaptic signaling in the GO Biological Processes database and for synaptic membranes in the GO Cellular Components database. This highlights the biological specificity of TC121 in regulating gene expression linked to synaptic functions.

## Distinct cluster of patients from TCGA overlaps with elevated activity of TC121

To explore if pre-existing classifications of patients with ovarian cancer correspond to the contrasting activities of the TCs, we investigated the classification provided by TCGA. TCGA identified four optimal clusters describing the patients with ovarian cancer using transcriptional profiles (*Bell et al., 2011*). To explore associations between these clusters and TC activity, we performed a Kruskal-Wallis test using TCGA sample data. *Figure 5—figure supplement 3* highlights the associations between each cluster set and the TCs, represented by log-transformed p-values. A significant association between a TC and a cluster set indicates that at least one cluster within the cluster set exhibited significantly different activity scores for the corresponding TC compared to the other clusters. Notably, samples with high TC121 activity were not captured by any of the clusters of the four-cluster set. Interestingly, the eight-cluster set predefined by TCGA was able to identify a cluster that corresponded to samples with elevated TC121 and TC146 activity. This finding suggests that while TCGA's analysis identified this patient group based on transcriptional profiles, it didn't characterize them further.

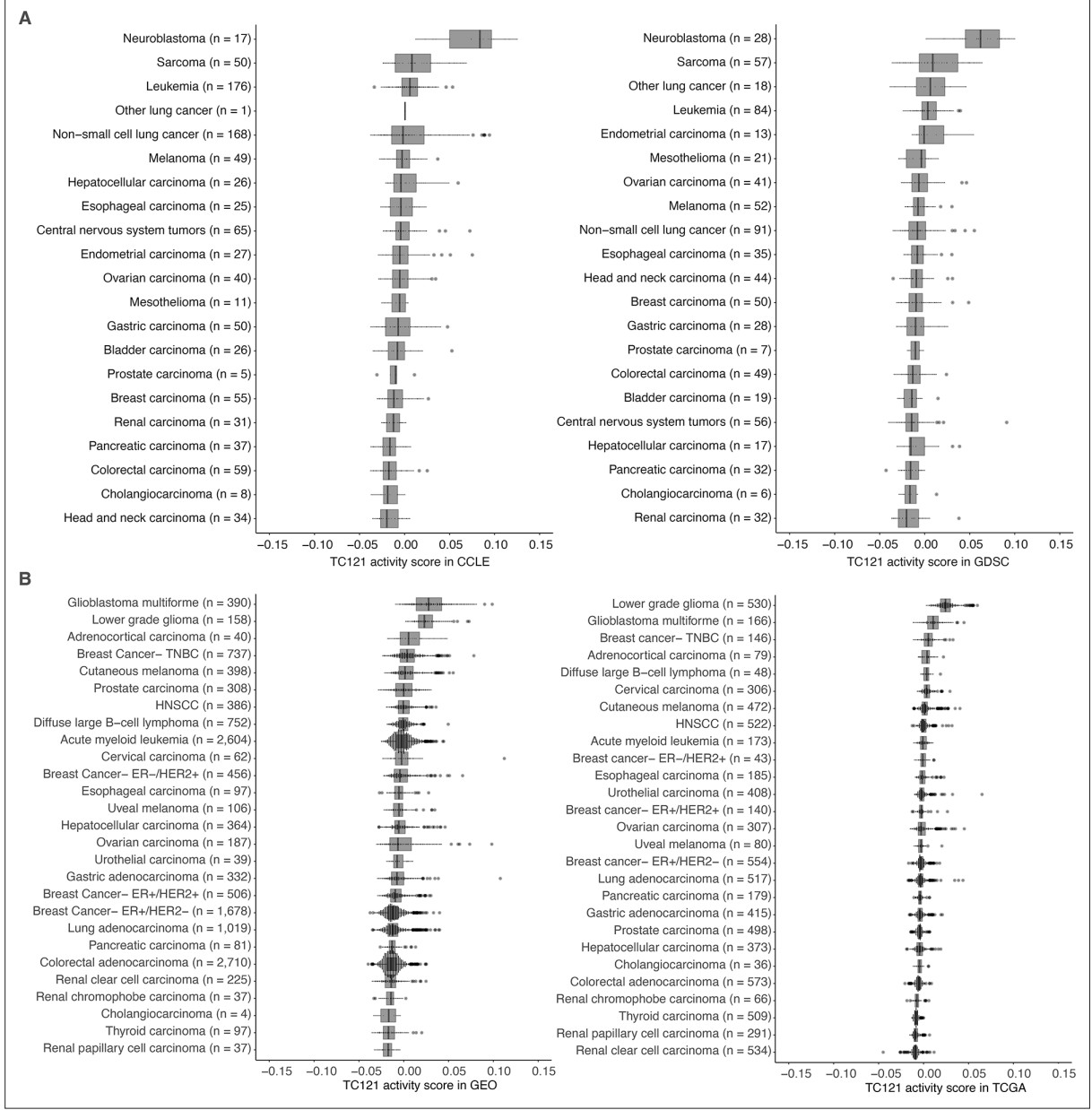

**Figure 5.** The activity of TC121 in bulk transcriptomes of Cancer Cell Line Encyclopedia (CCLE), Genomics of Drug Sensitivity in Cancer (GDSC) cell lines, and Gene Expression Omnibus (GEO) and The Cancer Genome Atlas (TCGA) patient-derived samples. (**A**) Cross-study TC projection of TC121 on CCLE and GDSC cell lines. The boxplots display the activity scores of TC121 in different tissue types, which are ordered based on their corresponding median activity scores. (**B**) Cross-study TC projection of TC121 on GEO and TCGA bulk transcriptomes resulted in the activity scores presented in the boxplots. Cancer types were ordered based on corresponding medians of TC121 activity scores. Abbreviations: TC = transcriptional component.

The online version of this article includes the following figure supplement(s) for figure 5:

**Figure supplement 1.** Activity of overall survival (OS)-associated TCs in bulk transcriptomes of Cancer Cell Line Encyclopedia (CCLE) cell lines.

**Figure supplement 2.** The activity of overall survival (OS)-associated TCs in bulk transcriptomes of Genomics of Drug Sensitivity in Cancer (GDSC) cell lines.

**Figure supplement 3.** Association heatmap for The Cancer Genome Atlas (TCGA) cluster sets with the activity scores of OS-related TCs.

## Distinct spatial and single-cell transcriptional profiles with high activity of OS-associated TCs

Cross-study TC projection onto spatial transcriptomic profiles from 11 ovarian cancer samples revealed that TC121 was highly active in profiles from the tumor region of the 11 ovarian cancer

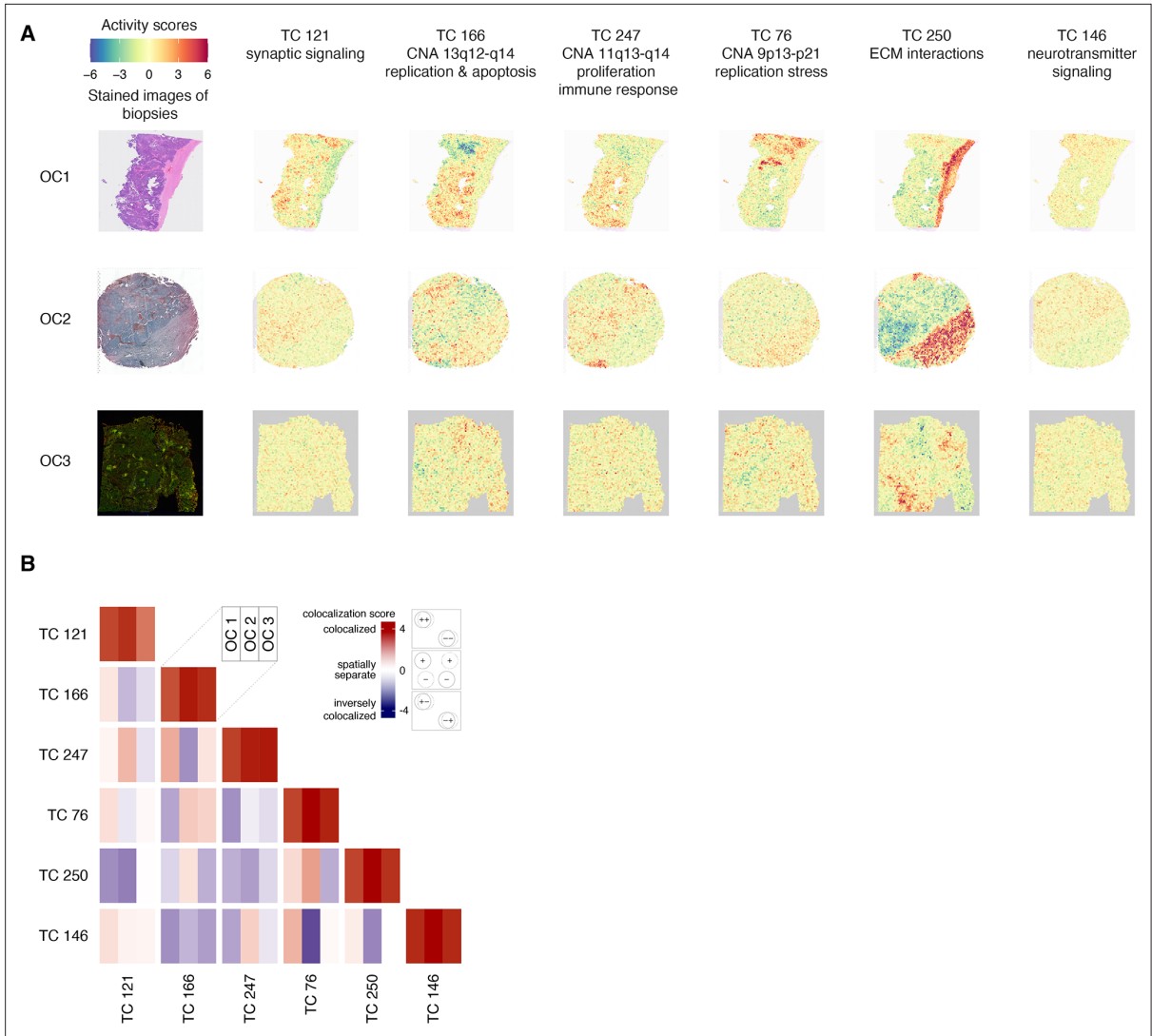

**Figure 6.** Spatial transcriptomic profiles in ovarian cancer samples. (**A**) We employed a permutation-based approach to pinpoint the areas of significant TC activity in spatial transcriptomic profiles. We ran 5000 permutations for each TC-profile combination, yielding a p-value that indicates the extent to which the TC activity in the corresponding profile differs from what would be expected by chance (the null distribution). We then transformed these p-values into logarithmic values and represented them using a heatmap. Heatmaps of activity scores of the TCs are presented in individual rows for the high-grade serous ovarian cancer (HGSOC), serous papillary, and endometrioid adenocarcinoma of ovary samples. The first column represents the stained images of the samples. The second to seventh columns show heatmaps corresponding to the mentioned TCs. (**B**) The heatmap illustrates the colocalization between two TC activities on spatial transcriptomic profiles from ovarian cancer samples. For each cell, the colocalization scores of the TCs at each of the three spatial transcriptomics samples OC 1, OC 2, and OC 3 are arranged in columns. A colocalization score of 4 between two TCs (red) indicates that the positively (+) and negatively (-) active regions of both TCs are perfectly colocalized. Conversely, a colocalization score of –4 between two TCs (blue) also indicates colocalization. Still, with inverse activity, i.e., the positively active regions of one TC are colocalized with the negatively active regions of the other TC or vice versa. A colocalization score close to 0 between two TCs (white) indicates that the activities of two TCs are spatially separated. The dashed and solid circles in the panel on the right side of the color bar represent two different TCs. Abbreviations: TC = transcriptional component.

The online version of this article includes the following figure supplement(s) for figure 6:

**Figure supplement 1.** Spatial transcriptomic profiles in eight ovarian cancer samples.

**Figure supplement 2.** The activity scores of OS-associated TCs in different single-cell types from patients with HGSOC.

samples (*Figure 6A*; *Figure 6—figure supplement 1*). Additionally, TC121 showed markedly higher activity in the transcriptional profiles of a subset of the unannotated single cells from HGSOC patients (study ID GSE158722; see supplementary methods and *Figure 6—figure supplement 2*).(*Nath et al., 2021*) This finding suggests that some of these unannotated cells could be neurons. Furthermore, the

unannotated single-cell transcriptional profiles showed contrasting activity scores of different OS-associated TCs (*Figure 6—figure supplement 2*). These contrasting activities indicate that these TCs could provide insights into the biology of previously uncharacterized cell types. Distinct regions with high activity of the copy number TCs (TC166, TC247, and TC76) in the HGSOC sample overlapped with the region containing cancer cells, as expected. TC250, enriched for extracellular matrix interactions, was also active in the stromal region. The strongest inverse colocalization (colocalization score –2.43) was observed between the activity scores of TC146, enriched for neurotransmitter signaling, and TC76, which captured the effect of copy number alterations at chromosome 9p13-p21, at the serous ovarian cancer sample (*Figure 6B*, *Supplementary file 10*).

## Discussion

In this study, we identified 374 TCs, each enriched for gene sets representing various biological processes in HGSOC samples. Six could stratify patients with HGSOC who had received platinum-based treatment into ten distinct OS groups.

The most significant TC in the survival tree analysis, TC121, captured a clinically relevant subtle transcriptional pattern linked to synaptic signaling not previously recognized in HGSOC. In the survival tree, TC121 identified 12% of the HGSOC patients with the shortest OS and, based on spatially resolved transcriptomic analyzed samples, is active in tumor regions. This observation supports the emerging role of neurons and neuronal projections as cancer hallmark-inducing constituents of the TME (*Hanahan and Weinberg, 2011*; *Reavis et al., 2020*; *Gysler and Drapkin, 2021*).

Further investigation on whether the activity of TC121 originated from tumor cells or in the TME revealed that the TC121 signal is coming from cells within the TME. The high activity of TC121 in low-grade glioma and glioblastoma multiforme patient samples (*Figure 5B*) is in agreement with the presence of neurons in large numbers within the TME of gliomas, where they form functional synapses with tumor cells (*Radin and Tsirka, 2020*; *Venkatesh et al., 2019*). Moreover, TC121 activity was lower in non-brain cancers, such as ovarian cancers, which contain fewer neurons and synapses in the TME compared to brain cancers. We expected TC121 activity to be low in the bulk transcriptomes of all cell lines, since they lack TME. TC121 activity in most cell lines, which includes glioblastoma and ovarian cancer cell lines, was indeed low. Neuroblastoma cell lines, however, exhibited high TC121 activity, which is likely due to retained synaptic formation capacity originating from neuroblast cells (*De Preter et al., 2006*; *Mark et al., 2021*). Lastly, TC121's high activity observed in small, scattered regions within the tumor of spatially resolved transcriptomic ovarian cancer samples also supports TC121's role in the TME.

TC121's significant association with OS underscores the potential significance of synaptic signaling in HGSOC biology. Yet, the neuronal subtype and the molecular mechanisms associated with TC121 remain to be elucidated. A study in human ovarian cancer-bearing mice demonstrated that sympathetic innervation in HGSOC involves adrenergic signaling: norepinephrine released by sympathetic neurons binds to beta-adrenergic receptors on the cancer cells (*Allen et al., 2018*; *Rains et al., 2017*; *Eng et al., 2014*). This binding triggers the tumor cells to release brain-derived neurotrophic factor (BDNF), which enhances cancer innervation via activation of host neurotrophic receptor tyrosine kinase B receptors (NTRK2), thereby establishing a feed-forward loop of sustained signaling. BDNF and the nerve marker neurofilament protein expression were examined in 108 human ovarian tumors (*De Preter et al., 2006*). This study revealed that increased intratumoral nerve presence strongly correlates with elevated BDNF and norepinephrine levels, advanced tumor stage, and shorter OS in patients with ovarian cancer. This interaction can be targeted with pan-TRK inhibitors such as entrectinib and larotrectinib. Both drugs are showing promising results in multiple phase II trials, including ovarian cancer and breast cancer patients. Furthermore, a TRKB-specific inhibitor was developed (ANA-12), but has not been subjected to any clinical trials in cancer so far (*Drilon et al., 2017*; *Ardini et al., 2016*; *Drilon et al., 2018*; *Burris et al., 2015*). Our analysis indicated that *BDNF* is a prominent gene (with an absolute weight >3) in 10 TCs but not in TC121, suggesting that TC121 may indicate a distinct process unrelated to BDNF.

The significance of sensory innervation in HGSOC was evidenced by the co-localization of TRPV1, a marker for sensory neurons, and β-III tubulin, a general neuronal marker, in immunofluorescent staining of histological sections from 75 patients (*Barr et al., 2021*). Additionally, a murine model study employing neural tracing identified sensory neurons originating from local dorsal root ganglia

and jugular–nodose ganglia, with axons extending into the TME (*Barr et al., 2021*). A transgenic murine model lacking nociceptors demonstrated that this specific subtype of sensory neurons was involved in tumor progression (*Restaino et al., 2023*). Another study showed that reducing the release of calcitonin gene-related peptide from tumor-innervating nociceptors could be a strategy to alleviate this effect of nociceptors by improving anti-tumor immunity of cytotoxic CD8 + T cells in a melanoma model bearing mice (*Balood et al., 2022*). This indicates that the signal from TC121 may represent an indirect influence on tumor cells via interactions with immune cells and the promotion of an immune suppressive TME. Furthermore, in cell lines derived from Trp53$^{-/-}$ Pten$^{-/-}$ murine HGSOC, the influence of nociceptors was characterized by the release of substance P (SP), their primary neuropeptide. SP is an alternative splicing product of the preprotachykinin A gene (*TAC1*) and binds to the receptor neurokinin 1 (NK1R), encoded by the *TACR1* gene. NK1R expression was confirmed in the murine HGSOC cell line, and SP enhanced cellular proliferation in NK1R-positive murine HGSOC cancer cells in vitro (*Restaino et al., 2023*). Our analysis identified *TAC1* and *TACR1* as prominent genes in 15 and 2 TCs, respectively, yet not in TC121, and none of these TCs were associated with patient survival. Currently, there are no drugs specifically targeting tumor innervation in (ovarian) cancer (*Li et al., 2022*). Interestingly, the NK1R antagonist aprepitant effectively inhibited the metastasis-promoting effects of neural substance P in human breast and mammary cancer-bearing mice (*Padmanaban et al., 2024*), demonstrating the feasibility of such an approach. Strategies to disrupt neuronal signaling and neurotransmitter release in neurons target key elements of excitatory neurotransmission, such as calcium flux and vesicle formation. Drugs like ifenprodil and lamotrigine, commonly used to treat neuronal disorders, block glutamate release and subsequent neuronal signaling. Additionally, the vesicular monoamine transporter (VMAT) inhibitor reserpine prevents synaptic vesicle formation (*Williams, 2001*; *Reid et al., 2013*). In vitro studies with HGSOC cell lines have demonstrated that ifenprodil significantly inhibits tumor proliferation, while reserpine induces apoptosis in cancer cells (*Ramamoorthy et al., 2019*; *North et al., 2015*). These approaches hold promise for inhibiting neuronal signaling and interactions in the TME. Therefore, it is essential that the mechanisms driving this nerve growth, the specifics of how nerves within the TME interact with ovarian cancer cells, and how they impact the survival of patients with HGSOC are further elucidated.

Altogether, the present study uncovered a clinically relevant TC linked to synaptic signaling not previously identified in HGSOC. This TC may represent a novel cancer cell-extrinsic mechanism within the TME, illustrating how cancer cells and nerve cells interact to promote enhanced proliferation. A deeper understanding of the molecular aspects of tumor innervation could pave the way for novel drug targets for patients with HGSOC.

## Additional information

### Funding

| Funder | Grant reference number | Author |
| --- | --- | --- |
| Hanarth Fonds | 2019N1552 | Rudolf SN Fehrmann |

The funders had no role in study design, data collection and interpretation, or the decision to submit the work for publication.

### Author contributions

Arkajyoti Bhattacharya, Resources, Data curation, Software, Formal analysis, Supervision, Validation, Investigation, Visualization, Methodology, Writing – original draft, Project administration, Writing – review and editing; Thijs S Stutvoet, Data curation, Formal analysis, Investigation, Visualization, Writing – original draft, Writing – review and editing; Mirela Perla, Investigation, Visualization, Writing – original draft, Writing – review and editing; Stefan Loipfinger, Data curation, Formal analysis, Investigation, Visualization, Methodology, Writing – original draft, Writing – review and editing; Mathilde Jalving, Anna KL Reyners, Paola D Vermeer, Ronny Drapkin, Marco de Bruyn, Investigation, Writing – original draft, Writing – review and editing; Elisabeth GE de Vries, Supervision, Investigation, Visualization, Writing – original draft, Project administration, Writing – review and editing; Steven de Jong, Conceptualization, Supervision, Investigation, Visualization, Methodology, Writing – original

draft, Project administration, Writing – review and editing; Rudolf SN Fehrmann, Conceptualization, Resources, Data curation, Software, Formal analysis, Supervision, Funding acquisition, Validation, Investigation, Visualization, Methodology, Writing – original draft, Project administration, Writing – review and editing

## Author ORCIDs
Arkajyoti Bhattacharya (iD) https://orcid.org/0000-0003-1479-0739
Stefan Loipfinger (iD) https://orcid.org/0000-0002-5571-0435
Paola D Vermeer (iD) https://orcid.org/0000-0003-2370-8223
Marco de Bruyn (iD) https://orcid.org/0000-0001-9819-9131
Rudolf SN Fehrmann (iD) http://orcid.org/0000-0002-7516-315X

Reviewer #1 (Public review): https://doi.org/10.7554/eLife.101369.3.sa1
Reviewer #2 (Public review): https://doi.org/10.7554/eLife.101369.3.sa2
Author response https://doi.org/10.7554/eLife.101369.3.sa3

---

# Additional files

## Supplementary files
Supplementary file 1. Patient characteristics. Abbreviations: NA = not applicable, HGSOC = high-grade serous ovarian cancer, LGSOC = low-grade serous ovarian cancer, LMP = low malignant potential, OCCC = ovarian clear cell cancer.

Supplementary file 2. Overview of the number of samples from each Gene Expression Omnibus (GEO) series and the corresponding study. Abbreviations: PMID = PubMed identification number.

Supplementary file 3. Multivariate permutation framework results from the univariate survival analysis. Abbreviations: TC = transcriptional component. CI = confidence interval.

Supplementary file 4. Permutation framework results from the multivariate survival analysis. Abbreviations: TC = transcriptional component. CI = confidence interval.

Supplementary file 5. Clinicopathological parameters per survival tree node in platinum-treated HGSOC. Abbreviations: HGSOC = high-grade serous ovarian cancer, TC = transcriptional component, ECM = extracellular matrix.

Supplementary file 6. Clinicopathological parameters per survival tree node in advanced-stage platinum-treated HGSOC. Abbreviations: HGSOC = high-grade serous ovarian cancer, TC = transcriptional component, ECM = extracellular matrix.

Supplementary file 7. GenetICA gene network analysis results. Abbreviations: TC = transcriptional component.

Supplementary file 8. Results from the univariate survival analysis on patients with ovarian clear cell carcinoma. Abbreviations: TC = transcriptional component. CI = confidence interval.

Supplementary file 9. Expression and function of the top 20 genes in TC121, according to literature. Abbreviations: TC = transcriptional component.

Supplementary file 10. Colocalization scores of every two TC combinations in spatial transcriptomic profiles from three ovarian cancer samples. Abbreviations: TC = transcriptional component.

MDAR checklist

## Data availability
The current manuscript is a computational study, so no data have been generated for this manuscript. Microarray expression data was collected from three public data repositories: Gene Expression Omnibus with accession number GPL570 (generated with Affymetrix HG-U133 Plus 2.0), CCLE (generated with Affymetrix HG-U133 Plus 2.0, file CCLE_Expression.Arrays_2013-03-18.tar.gz) available at https://portals.broadinstitute.org/ccle/data and GDSC (generated with Affymetrix HG-U219) available at https://www.ebi.ac.uk/arrayexpress/experiments/E-MTAB-3610/. Pre-processed and normalized RNA-seq data was collected from TCGA using the Broad GDAC Firehose portal (https://gdac.broadinstitute.org/). Spatially resolved samples were sourced from 10xGenomics and GEO. The datasets generated during and/or analyzed during the current study are available in the website: https://transcriptional-landscape-ovarian.opendatainscience.net/. The complete set of codes utilized in this

study is available at the github repository: https://github.com/arkajyotibhattacharya/TranscriptionalLandscapeOvarianCancer (copy archived at *Bhattacharya, 2024*).

The following previously published datasets were used:

| Author(s) | Year | Dataset title | Dataset URL | Database and Identifier |
|---|---|---|---|---|
| Noh K, Birrer MJ, Sood AK | 2017 | Expression data from endothelial cell | https://www.ncbi.nlm.nih.gov/geo/query/acc.cgi?acc=GSE105437 | NCBI Gene Expression Omnibus, GSE105437 |
| Curry E | 2018 | Bivalent chromatin domains in ovarian tumours at initial presentation identify genes predisposed to DNA hypermethylation during acquired resistance to chemotherapy | https://www.ncbi.nlm.nih.gov/geo/query/acc.cgi?acc=GSE107931 | NCBI Gene Expression Omnibus, GSE107931 |
| Tone AA, Begley H, Sharma M, Murphy J, Rosen B, Brown TJ, Shaw PA | 2008 | Gene expression data from non-malignant fallopian tube epithelium and high grade serous carcinoma | https://www.ncbi.nlm.nih.gov/geo/query/acc.cgi?acc=GSE10971 | NCBI Gene Expression Omnibus, GSE10971 |
| Yeung T, Mok SC, Wong ST | 2019 | Systematic Identification of Epithelial-Stromal Crosstalk Signaling Networks in Ovarian Cancer | https://www.ncbi.nlm.nih.gov/geo/query/acc.cgi?acc=GSE115635 | NCBI Gene Expression Omnibus, GSE115635 |
| Anglesio MS, Arnold JM, George J, Tinker AV, Tothill R, Wadell N, Simms L, Locandro B, Fereday S, Traficante N, Russell P, Sharma R, Birrer MJ, deFazio A, Chenevix-Trench G, Bowtell DD | 2008 | Common activation of RAS_MAPK pathway in serous LMP tumours | https://www.ncbi.nlm.nih.gov/geo/query/acc.cgi?acc=GSE12172 | NCBI Gene Expression Omnibus, GSE12172 |
| Tung CS, Mok SC, Tsang YT, Zu Z, Liu J, Deavers MT, Malpica A, Wolf J | 2009 | PAX2: A Potential Biomarker for Low Malignant Potential Ovarian Tumors and Low-Grade Serous Ovarian Carcinomas | https://www.ncbi.nlm.nih.gov/geo/query/acc.cgi?acc=GSE14001 | NCBI Gene Expression Omnibus, GSE14001 |
| Bowen NJ, Walker LD, Matyunina LV, Totten K, Benigno BB, McDonald JF | 2009 | Ovarian Cancer gene expression profiling identifies the surface of the ovary as a stem cell niche | https://www.ncbi.nlm.nih.gov/geo/query/acc.cgi?acc=GSE14407 | NCBI Gene Expression Omnibus, GSE14407 |
| Pejovic T | 2009 | Expression Data from Ovarian Surface Kinome | https://www.ncbi.nlm.nih.gov/geo/query/acc.cgi?acc=GSE15578 | NCBI Gene Expression Omnibus, GSE15578 |
| Mok SC, Bonome T, Vathipadiekal V, Bell A | 2009 | A gene signature predictive for outcome in advanced ovarian cancer identifies a novel survival factor: MAGP2 | https://www.ncbi.nlm.nih.gov/geo/query/acc.cgi?acc=GSE18521 | NCBI Gene Expression Omnibus, GSE18521 |
| Iorio E, Ricci A, Bagnoli M, Pisanu ME | 2010 | Activation of phosphatidylcholine-cycle enzymes in human epithelial ovarian cancer cells | https://www.ncbi.nlm.nih.gov/geo/query/acc.cgi?acc=GSE19352 | NCBI Gene Expression Omnibus, GSE19352 |
| Konstantinopoulos PA, Spentzos D, Karlan BY, Taniguchi T | 2010 | A gene expression profile of BRCAness that is associated with outcome in ovarian cancer | https://www.ncbi.nlm.nih.gov/geo/query/acc.cgi?acc=GSE19829 | NCBI Gene Expression Omnibus, GSE19829 |

*Continued on next page*

*Continued*

| Author(s) | Year | Dataset title | Dataset URL | Database and Identifier |
|---|---|---|---|---|
| Meyniel J, Cottu PH, Decraene C, Stern M, Couturier J, Lebigot I, Nicolas A, Weber N, Fourchotte V, Alran S, Rapinat A, Gentien D, Roman-Roman S, Mignot L, Sastre-Garau X | 2010 | Primary and secondary ovarian tumors | https://www.ncbi.nlm.nih.gov/geo/query/acc.cgi?acc=GSE20565 | NCBI Gene Expression Omnibus, GSE20565 |
| Consortium International Genomics | 2019 | Expression Project for Oncology (expO) | https://www.ncbi.nlm.nih.gov/geo/query/acc.cgi?acc=GSE2109 | NCBI Gene Expression Omnibus, GSE2109 |
| Mateescu B, Batista L, Mariani O, Meyniel J, Cottu PH, Sastre-Garau X, Mechta-Grigoriou F | 2011 | Control of oxidative stress by miRNA and impact on ovarian tumorigenesis | https://www.ncbi.nlm.nih.gov/geo/query/acc.cgi?acc=GSE26193 | NCBI Gene Expression Omnibus, GSE26193 |
| King ER, Tung CS, Tsang YT, Zu Z | 2011 | The Anterior Gradient Homolog 3 (AGR3) Gene Is Associated with Differentiation and Survival in Ovarian Cancer | https://www.ncbi.nlm.nih.gov/geo/query/acc.cgi?acc=GSE27651 | NCBI Gene Expression Omnibus, GSE27651 |
| Wong KK, Tsang YT, Deavers MT, Mok SC | 2010 | BRAF Mutation Is Rare in Advanced-Stage Low-Grade Ovarian Serous Carcinomas | https://www.ncbi.nlm.nih.gov/geo/query/acc.cgi?acc=GSE27659 | NCBI Gene Expression Omnibus, GSE27659 |
| Stany MP, Vathipadiekal V, Ozbun L, Stone RL | 2011 | Identification of Novel Therapeutic Targets in Microdissected Clear Cell Ovarian Cancers | https://www.ncbi.nlm.nih.gov/geo/query/acc.cgi?acc=GSE29450 | NCBI Gene Expression Omnibus, GSE29450 |
| Yoshihara K, Tsunoda T, Shigemizu D, Fujiwara H | 2012 | Immune-activation as a therapeutic direction for patients with high-risk ovarian cancer based on gene expression signature (1) | https://www.ncbi.nlm.nih.gov/geo/query/acc.cgi?acc=GSE32062 | NCBI Gene Expression Omnibus, GSE32062 |
| Roth RB, Hevezi P, Lee J, Willhite D | 2006 | Comparison of gene expression profiles across the normal human body | https://www.ncbi.nlm.nih.gov/geo/query/acc.cgi?acc=GSE3526 | NCBI Gene Expression Omnibus, GSE3526 |
| Elgaaen BV, Olstad OK, Sandvik L, Ødegaard E, Sauer T, Staff AC, Gautvik KM | 2012 | Expression data from serous ovarian carcinomas, serous ovarian borderline tumors and surface epithelium scrapings from normal ovaries | https://www.ncbi.nlm.nih.gov/geo/query/acc.cgi?acc=GSE36668 | NCBI Gene Expression Omnibus, GSE36668 |
| Abiko K, Mandai M | 2013 | Gene-expression profiles of ascites-cytology-positive ovarian cancer | https://www.ncbi.nlm.nih.gov/geo/query/acc.cgi?acc=GSE39204 | NCBI Gene Expression Omnibus, GSE39204 |
| Yeung TL, Leung CS, Wong KK, Samimi G | 2013 | A cancer associated fibroblasts (CAFs) specific gene signature in high grade serous ovarian cancer | https://www.ncbi.nlm.nih.gov/geo/query/acc.cgi?acc=GSE40595 | NCBI Gene Expression Omnibus, GSE40595 |
| Wu Y, Chang T, Huang Y, Huang H, Chou C | 2014 | COL11A1 promotes tumor progression and predicts poor clinical outcome in ovarian cancer | https://www.ncbi.nlm.nih.gov/geo/query/acc.cgi?acc=GSE44104 | NCBI Gene Expression Omnibus, GSE44104 |

*Continued on next page*

*Continued*

| Author(s) | Year | Dataset title | Dataset URL | Database and Identifier |
|---|---|---|---|---|
| Koti M, Goooding R, Squire JA | 2013 | Gene expression data from high grade serous ovarian cancer | https://www.ncbi.nlm.nih.gov/geo/query/acc.cgi?acc=GSE51373 | NCBI Gene Expression Omnibus, GSE51373 |
| Hill CG, Matyunina LV, Walker D, Benigno BB | 2014 | Transcriptional override: a regulatory network model of indirect responses to modulations in microRNA expression | https://www.ncbi.nlm.nih.gov/geo/query/acc.cgi?acc=GSE52460 | NCBI Gene Expression Omnibus, GSE52460 |
| Abiko K, Matsumura N | 2015 | Gene-expression profiles of ovarian cancer regarding its microenvironment | https://www.ncbi.nlm.nih.gov/geo/query/acc.cgi?acc=GSE55512 | NCBI Gene Expression Omnibus, GSE55512 |
| Lisowska K, Kupryjańczyk J | 2014 | Gene expression profiling in ovarian cancer | https://www.ncbi.nlm.nih.gov/geo/query/acc.cgi?acc=GSE63885 | NCBI Gene Expression Omnibus, GSE63885 |
| Uehara Y, Oda K, Ikeda Y, Koso T, Tsuji S, Yamamoto S, Asada K, Sone K, Kurikawa R, Sone K, Makii C, Hagiwara O, Tanikawa M, Maeda D, Hasegawa K, Nakagawa S, Wada-Hiraike O, Kawana K, Fukayama K, Yano T, Osuga Y, Fujii T, Aburatani H | 2015 | Integrated copy number and expression analysis identifies profiles of whole-arm chromosomal alterations and subgroups with favorable outcome in ovarian clear cell carcinomas | https://www.ncbi.nlm.nih.gov/geo/query/acc.cgi?acc=GSE65986 | NCBI Gene Expression Omnibus, GSE65986 |
| Yamamoto Y, Ning G, Mehra K, Tay A, McKeon F, Crum CP, Xian W | 2016 | Transformation of Human Fallopian Tube Stem Cells and high grade serous ovarian cancer | https://www.ncbi.nlm.nih.gov/geo/query/acc.cgi?acc=GSE69428 | NCBI Gene Expression Omnibus, GSE69428 |
| Gao Q, Yang Z, Xu S, Li X | 2019 | Gene expression profile of tumor cells from primary tumors, ascites and metastases of high and low grade serous ovarian cancer patients | https://www.ncbi.nlm.nih.gov/geo/query/acc.cgi?acc=GSE73168 | NCBI Gene Expression Omnibus, GSE73168 |
| Tothill RW, Tinker AV, George J, Brown R | 2008 | Expression profile of ovarian tumour samples | https://www.ncbi.nlm.nih.gov/geo/query/acc.cgi?acc=GSE9899 | NCBI Gene Expression Omnibus, GSE9899 |
| Iorio F | 2022 | Transcriptional Profiling of 1,000 human cancer cell lines | https://www.ebi.ac.uk/arrayexpress/experiments/E-MTAB-3610/ | ArrayExpress, E-MTAB-3610 |
| Affymetrix Inc | 2003 | [HG-U133_Plus_2] Affymetrix Human Genome U133 Plus 2.0 Array | https://www.ncbi.nlm.nih.gov/geo/query/acc.cgi?acc=GPL570 | NCBI Gene Expression Omnibus, GPL570 |

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

# Appendix 1

## Supplementary methods

### Data acquisition

Publicly available raw microarray bulk transcriptomics were extracted from the GEO (*Clough and Barrett, 2016*). We are confined to the Affymetrix HG-U133 Plus 2.0 platform (GEO accession identifier: GPL570). Samples were selected for analysis if they represented epithelial ovarian cancer (EOC) obtained from patients. Also, samples were collected from non-cancerous ovarian tissue obtained with surface brushings or laser micro-dissected ovarian epithelium (n=25), ovarian epithelial cells isolated using short-term culture (n=9), or derived from complete ovaries (n=9). Cell line samples were excluded. For all selected samples, relevant clinicopathological data were collected whenever available, including grade, stage, age, treatment history, debulking status, progression-free survival (PFS), and OS.

### CCLE dataset

Raw bulk transcriptomic data was obtained from the CCLE project, which conducted a detailed genetic characterization of a large panel of human cancer cell lines (*Barretina et al., 2012*). Expression data within the CCLE project was generated with Affymetrix HG-U133 Plus 2.0. This dataset is referred to as the CCLE dataset throughout this manuscript.

### GDSC dataset

From the GDSC portal, we obtained raw expression data generated with Affymetrix HG-U219 (*Yang et al., 2013*). The aim of the GDSC project is to identify molecular features of cancer that predict response to anti-cancer drugs. This dataset is referred to as the GDSC dataset throughout this manuscript.

### TCGA dataset

From TCGA, we obtained the pre-processed and normalized level 3 RNA-seq (version 2) data for 27 cancer datasets available at the Broad GDAC Firehose portal (downloaded January 2017 https://gdac.broadinstitute.org/). For each sample, we downloaded RNA-Seq with Expectation Maximization (RSEM) gene normalized data (identifier: illuminahiseq_rnaseqv2 RSEM_genes_normalized) (*Broad Institute TCGA Genome Data Analysis Center, 2016*).

### Spatial transcriptomic profiles

From the 10x Genomics repository, we sourced the spots-without-tissue-filtered Spatial transcriptomic profiles of three samples from ovarian cancer patients in h5 format. Stained images of these three samples along with scale factors and tissue positions were also downloaded from the same repository. These three samples were collected from patients who had high-grade serous ovarian cancer, serous papillary carcinoma, and endometrioid adenocarcinoma of the ovary respectively. The repositories are mentioned in the following:

1. https://www.10xgenomics.com/datasets/human-ovarian-cancer-11-mm-capture-area-ffpe-2-standard
2. https://www.10xgenomics.com/datasets/human-ovarian-cancer-1-standard
3. https://www.10xgenomics.com/datasets/human-ovarian-cancer-whole-transcriptome-analysis-stains-dapi-anti-pan-ck-anti-cd-45-1-standard-1-2-0

We also collected publicly available spatial resolved transcriptomic profiles from eight ovarian cancer samples sourced from GEO (study ID GSE211956).

### Sample processing and quality control

Non-corrupted raw data CEL files were downloaded from GEO for the selected samples. To identify samples that were uploaded to GEO multiple times, we generated an MD5 hash for each CEL file. After the removal of duplicate CEL files, preprocessing and aggregation of CEL files was performed with robust multiarray averaging (RMA using the justRMA function from R package aroma.affymetrix v3.2.0) method using R version 3.5.2. Quality control was performed using principal component analysis as previously described (*Bhattacharya et al., 2020*). MD5 hash duplicate removal does not detect identical samples when meta-data is different. Pearson correlation coefficients among bulk transcriptomics were obtained to identify samples with identical expression values. One sample from

each set of duplicate samples was randomly chosen, and the rest were removed from subsequent analyses.

## Consensus-independent component analysis (c-ICA)

The bulk transcriptomics included in this analysis were generated with complex tumor tissues. These tissues contain a complex mixture of heterogeneous tumor cells and non-tumor cells present in the tumor microenvironment. Therefore, the resulting profiles represent the average transcriptomic patterns of cells present in the biopsies. c-ICA was utilized to segregate the average transcriptomic patterns of complex biopsies into statistically independent transcriptional components (*Chiappetta et al., 2004*).

Applying ICA on a bulk transcriptomic dataset with $p$ genes and $n$ samples results in the extraction of $i$ independent components of dimension $1 \times p$ (hereafter called estimated sources, ESs) and a mixing matrix (MM) of dimension $i \times n$ which contains the coefficients of ESs in each sample. The weight of each ES represents the direction and magnitude of its effect on the expression level of each gene, and the coefficients of MM represent the activity scores of the ESs in the corresponding sample. In ICA, a preprocessing technique called whitening is applied to the input dataset to make the estimation more time-efficient. Whitening was used to transform bulk transcriptomics of all samples so that the transformed profiles are uncorrelated and have a variance of one. Next, ICA was performed on the whitened dataset using the FastICA function from the FastICA package (version 1.2.0), resulting in the extraction of $i$ independent components and a mixing matrix. The parameter $i$ was chosen as the number of top principal components, which captured 90% of the total variance seen in the whitened dataset.

In ICA, an initial random weight vector with a variance of 1 has to be chosen to obtain statistically independent ESs. Hence, different initial random weight vectors could result in different sets of ESs. To retrieve a set of consensus ESs (hereafter called transcriptional components, TCs), we performed 25 ICA runs, each with a different random initialization weight vector. The assumption is that over a large number of runs of ICA, the fastICA algorithm does not converge to any local solution for most of the runs. ESs extracted from these runs were clustered together if the absolute value of the Pearson correlation between them was >0.9. Clusters with ESs from >50% of the runs were used to obtain $m$ TCs using the following formula:

$$TC_{p \times 1} = (1/n) \sum_{i=1}^{n} \left( ES_i \times sign \left( correlation \left( ES_1, ES_i \right) \right) \right)$$

These $m$ TCs and the bulk transcriptomic dataset with $p$ genes and $n$ samples ($X_{p \times n}$) was used in the below formula to obtain a consensus mixing matrix (or CMM) which contains the coefficients of TCs in each sample:

$$CMM_{m \times n} = \left( \left( TC' \right)_{m \times p} \times TC_{p \times m} \right)^{-1} \times \left( TC' \right)_{m \times p} \times X_{p \times n}$$

Similar to the MM, the coefficients of CMM represent the activity scores of the TCs in the corresponding sample.

## Univariate survival analysis

As the coefficients of CMM for each TC vary from one sample to another, we hypothesized that these activity scores of the TCs might be associated with OS. To assess this association, we calculated the $-\log_{10}$(p-value) from Cox regression for each $TC_i$ as the predictor, denoted as "original_minus_log10_$p_i$." To reduce the probability of false-positive association due to multiple testing, we conducted a permutation test. First, the sample labels of the TC activities in the MM were permuted 10,000 times to ensure that the correlation structure remained intact, resulting in 10,000 sets of permuted TCs. Then, on each $TC_i$ separately, the following steps were conducted:

1. Cox regression is performed using the permutated $TC_i$ activities.
2. From the analyses of step 1, -log10(p-value) corresponding to all permuted TCs are obtained (permuted_minus_log10_p).
3. Sort the values of permuted_minus_log10_p in decreasing order (sp_minus_log10_p).

4. Sort the values of original_minus_log10_p in decreasing order (so_minus_log10_p).
5. For every $j$-th value of so_minus_log10_p (so_minus_log10_$p_j$), obtain the number of values of sp_minus_log10_$p_i$ greater than so_minus_log10_$p_j$, defined as $f_j$.
6. We subset so_minus_log10_p till the j-th entry where $f_j$ / j>1% (false discovery rate of 1%).
7. Obtain the optimal cutoff for defining false discoveries of sp_minus_log10_$p_i$ ($oc_i$) as the maximum value of so_minus_log10_p.

Next, we defined a single optimal cutoff of $oc$ from all the TCs, as an 80% quantile of values of occ (confidence level of 80%). For each $TC_i$, if original_minus_log10_$p_i$ ≥ the single optimal cutoff, it was considered statistically significantly associated with OS in the permutation test framework.

## Multivariate survival analysis

Next, we performed multivariate Cox proportional hazards analysis using the known prognostic parameters age, stage, debulking status, and grade along with the TCs one at a time as predictors. A smaller subset of samples containing information about all of these clinicopathological variables was used for this analysis. For each $TC_i$, multivariate survival analysis with OS as the outcome variable was conducted using the following steps:

1. -$\log_{10}$(adjusted p-value) for each $TC_i$ as a predictor (original_minus_log10_adj_$p_i$) were obtained from the multivariate survival analysis.
2. Permutation tests were performed on each $TC_i$ exactly as described in the univariate survival analysis section. As a result, we obtained a list of TCs which are statistically significantly associated with OS after removing the effects of age, stage, debulking status, and grade in the permutation test framework.

## Survival tree analysis

Survival tree analysis is used to classify samples with clinicopathological information based on the difference between survival probabilities at certain time intervals. A subset of platinum-treated SOC samples with follow-up information was used in the survival tree analysis. All TCs associated with OS in the univariate or multivariate survival analyses served as classifiers along with all clinicopathologic parameters (age, stage, debulking status, and grade). Survival tree analysis was performed using the following steps:

1. For each of the classifiers ($C_i$):
   1. If the classifier is a numeric variable,
      1. For each possible weight of the classifier as a cutoff (c):
         1. Obtain two subsets of the samples.
            1. Samples having weights <cutoff weight
            2. Or samples having weights ≥cutoff weight
         2. Obtain survival probabilities for both of the subsets.
         3. Compare the survival probabilities using the log-rank test and obtain the log-rank statistic ($LRC_{i,c}$).
      2. Obtain the optimum cutoff c ($oc_i$) for which log-rank statistic $LRC_{i,c}$ is maximum. Assign a maximum of $LRC_{i,c}$ as max_$LRC_i$.
   2. If the classifier is a categorical variable,
      1. For each possible combination of different levels of the classifier:
         1. Obtain two subsets of samples A and B.
         2. Obtain survival probabilities for both of the subsets.
         3. Compare the survival probabilities using the log-rank test and obtain the log-rank statistic ($LRC_{i,AB}$).
      2. Obtain the optimum combination of levels in two subsets A, B ($oc_i$) for which log-rank statistic $LRC_{i,AB}$ is maximum. Assign a maximum of $LRC_{i,AB}$ as max_$LRC_i$.
2. Obtain the most significant classifier along with the optimum cutoff/combination of levels (oc) for which max_$LRC_i$ is maximum among all classifiers.
3. Classify the samples into two subsets (subset_1 and subset_2) using the most significant classifier and optimum cutoff/combination of levels.
4. In each of the subsets of the dataset, repeat steps 1, 2, & 3 till the following constraints are maintained.
   1. Number of samples in subset_1+number of samples in subset_2≥50

2. Number of uncensored events in subset_1+number of uncensored events in subset_2≥25
3. Number of samples in subset_1 or number of samples in subset_2≥17

After that, 10,000 iterations of the above four steps were performed using random 80% of the samples each time to investigate the robustness of the significance of the classifiers obtained from the survival tree analysis mentioned above.

To assess the goodness-of-fit of the survival tree, the following steps were conducted:

1. A new variable called survival cohorts was created to store the terminal node number in the survival tree of each sample.
2. Univariate Cox regression with this new variable survival cohorts as a predictor was performed to assess its association with OS.
3. Concordance statistic for this Cox regression model was obtained to evaluate the classification power of the survival tree.

To quantify the robustness of the survival tree, the following steps were conducted:

1. Survival tree analysis was conducted 20,000 times using random 80% of the samples each time.
2. For the survival tree with all samples
   1. For each classifier $C$
      1. NOS_node$_{C,n}$ was obtained as the number of samples present in node $n$ where $C$ was the most significant classifier.
      2. Max_NOS_node$_C$ was obtained as the maximum of NOS_node$_{C,n}$.
      3. R_all_samples_survival_tree$_C$ was obtained as a rank of Max_NOS_node$_C$.
3. For each of the 20,000 iterations $i$
   1. For each classifier $C$
      1. NOS_node$_{C,n}$ was obtained as the number of samples present in node $n$ where $C$ was the most significant classifier.
      2. Max_NOS_node$_C$ was obtained as the maximum of NOS_node$_{C,n}$.
      3. R$_{C,i}$ was obtained as the rank of Max_NOS_node$_C$.
   2. Robustness_statistic$_i$ was obtained as the Spearman correlation coefficient between R$_{C,i}$ and R_all_samples_survival_tree$_C$

The robustness of the survival tree was presented by the interquartile range and median of Robustness_statistic$_i$.

## Associating the identified transcriptional components with biological processes

We characterized biological activity represented by a TC using multiple methods. First, based on our recently published method called Transcriptional Adaptation to Copy Number Alterations (TACNA) profiling, we identified a subset of TCs that capture the downstream effects of CNAs on mRNA expression levels[5]. Second, we performed GSEA on each TC using gene set collections (n=16) collected from The Human Phenotype Ontology (The Monarch Initiative), the Mammalian Phenotypes (Mouse Genome Database), and the Molecular Signatures Database (MsigDB) *Subramanian et al., 2005*; *Liberzon et al., 2011*; *Liberzon et al., 2015*. We included all gene sets with 10–500 genes after filtering out genes that were not present in the expression profiles. Enrichment of each gene set was tested according to the two-sample Welch's t-test for unequal variance between the set of genes, which were under investigation, versus the set of genes that was not under investigation. To allow comparison between gene sets of different sizes, we transformed the Welch's t statistic to a Z-score.

Finally, we used the genetICA-network (available at http://www.genetica-network.com), to predict the likely function of the top genes inside a component (*Urzúa-Traslaviña et al., 2021*). In short, a GBA approach was used to predict likely functions for genes based on gene co-regulation. For this, we conducted a consensus ICA on an unprecedented scale. In short, a covariance matrix was calculated between 19,635 genes using the expression patterns of 106,462 bulk transcriptomics generated with Affymetrix HG-U133 Plus 2.0 representing the many disease states, cellular states, and genetic and chemical perturbations that were obtained. Consensus-ICA was performed on the covariance matrix. This identified a large set of CESs and a mixing matrix reflecting the activity of each source in the expression pattern of the gene across the samples. Next, a GBA approach was used to predict the functionality of individual genes. First, we retrieved 16 public gene set collections describing a large range of biological processes and phenotypes. For each gene set, we calculated

its 'bar code' by averaging the MM weight of its member genes. Next, for each gene in the MM, the distance correlation was determined between its MM weights and the gene set bar code. A high correlation between a gene's MM weight and a gene set bar code indicated that the gene under investigation shared functionality with the genes of the specific gene set under investigation. Significance levels were obtained with permutated data (250 permutations). This strategy was used on 23,372 well-described functional gene sets, which enabled us to create a comprehensive network of predicted functionalities of individual genes. From each studied TC, we separately selected the top and bottom 250 genes with a weight of ≥3 or ≤ –3 and created co-functionality networks using a network threshold of 0.65. For all resulting gene clusters consisting of ≥5 genes, the top 10 gene sets with a mean Z-score >2 were used for interpretation of the biological signal within a TC.

### Cross-study transcriptional component projection

A cross-study transcriptional component projection was conducted to investigate the activity scores of the transcriptional components in independently obtained mRNA expression profiles. Following steps were conducted for each cross-study transcriptional component projection analysis where transcriptional components (TC) of dataset $i$ were used to obtain activity scores of the same TC's in samples of dataset $j$:

1. Genes not present in both datasets $i$ and $j$ were removed from the analysis.
2. Both dataset $i$ and dataset $j$ were standardized on gene level separately, which means each gene expression is transformed to a mean of zero and standard deviation of one.
3. Both of these standardized datasets were sample-wise merged to obtain a combined dataset (Combined_i_used_for_j).
4. Transcriptional component matrix of dataset $i$ ($TC_i$) and Combined_i_used_for_j were used to obtain a consensus mixing matrix (CMMcombined_i_used_for_j). CMMcombined_i_used_for_j $= \left( \left( TC_i \right)' \times TC_i \right)^{-1} \times \left( TC_i \right)' \times$ Combined_i_used_for_j
5. The subset of coefficients CMMcombined_i_used_for_j for samples from dataset j is obtained as CMM_j

### Identification of significant activity of each TC in spatial transcriptomic profiles

To identify if the activity scores of each TC in each spatial transcriptomic profile are significantly different from the null distribution of possible activity scores, we conducted the following steps:

- A set of 3000 permutations of all the gene weights in the TC i were conducted (i[th] permuted TC is denoted by *TC_permuted_i*). For each permutation r:

  a. Obtain the permuted activity score for the samples using the following formula:
  $CMM\_permuted_r$

  $$= \left( \left( TC\_permuted\_i' \right)_{1 \times p} \times TC\_permuted\_i_{p \times 1} \right)^{-1}$$
  $$\times \left( TC\_permuted\_i' \right)_{1 \times p} \times X_{p \times n}$$

  b. Thereafter, a Johnson transformation was conducted on the vector *CMM_permuted* so that the distribution was as similar as possible to normal distribution with mean zero and standard deviation of one (referred to as *CMM_permuted_Johnsontransformed*). The same transformation was applied to the original CMMi,s.

  c. Finally, a p-value was obtained based on the position of the Johnson transformed CMMi,s with respect to the generalized normal distribution fitted to the *CMM_permuted_Johnsontransformed*

  d. Log-transformed p-values were thereafter plotted in a heatmap incorporating the row and column position of the individual spatial transcriptomic profiles to visualize the location of significant activity of the TCs.

### Determination of spatial transcriptomic profiles' significant activity locations for individual transcriptional components

Public spatial resolved transcriptomic profiles from three ovarian cancer samples, generated using the 10xGenomics Visium platform, were obtained for analysis. The samples were from patients

with high-grade serous, serous papillary, and endometrioid ovarian cancer, respectively. Activity for each TC across every location within the spatial samples was ascertained through the cross-study projection methodology referred to in the previous method section. To discern the markedly active areas within the spatial samples for each TC, we incorporated a permutation-driven approach. We derived a null distribution of activities for each TC-location pairing by performing 3000 permutations and subsequent projection. The p-value of the observed activity was set as the number of permutations that gave a higher activity divided by the number of permutations. This p-value quantifies the significance of the deviation of the TC's activity at a given location from its baseline null distribution. After this, we visualized the p-values, and post their log transformation, using a heatmap. This visualization aided in highlighting the areas with notable activity, aligned against the stained representation of the tissue sample.

## Colocalization analysis of TC activities in spatial transcriptomics

The colocalization method is based on the principles of colocalization analysis in microscopy images and is adapted from a previously published colocalization analysis (*Grisanti Canozo et al., 2022*). We first chose highly (in)active regions by selecting spots where the TCs activity of the permutation-corrected mixing matrix of the spatial transcriptomic images was above 2 for highly active or below –2 for inactive TC processes. Then, for each image and TC, kernel density estimation was performed to estimate the density of highly active or inactive regions using the R package ks 1.14.1 (*Chacón and Duong, 2018*). Only TCs with at least 20 active or inactive spots were considered. The optimal kernel bandwidth was determined using the least-squares cross-validation bandwidth selector with the method "Hlscv.diag" and an initial bandwidth matrix of [(9,0),(0,9)]. The binned background grid (bgridsize) was defined as the dimensions of the spots in the spatial transcriptomic image. The kernel was then fitted using the "kde" function with weights being the absolute TC activity per spot of the (in)active regions. Next, we retained only those locations where the kernel density regions exceeded the 75$^{th}$ percentile of the kernel densities. We then assessed the similarity of kernel densities for all combinations of active and inactive regions of TCs of an image by taking the union of the selected regions and calculating the Pearson correlation coefficients $\rho$ between the density values of two TCs. A $\rho$ of 1 indicates that the (in)active regions of two TCs are colocalized, while a $\rho$ of –1 indicates that they are spatially exclusive. To enhance the colocalization analysis between TCs, we considered both coexisting and repelling properties of (in)active processes. To quantify this, we implemented a colocalization score between two TCs $a$ and $b$, and their respective active and inactive regions as follows:

$$\text{colocalization score}_{a,b} = (\rho_{a\_active,b\_active} - \rho_{a\_active,b\_inactive}) - (\rho_{a\_inactive,b\_active} - \rho_{a\_inactive,b\_inactive})$$

A colocalization score of 4 between two TCs indicates that the active and inactive regions of these TCs are colocalized, with the inactive regions of one TC being spatially exclusive to the active regions of the other TC and vice versa. Conversely, a colocalization score of –4 between two TCs also indicates colocalization, but with reversed activity, i.e., the active regions of one TC are colocalized with the inactive regions of the other TC or vice versa. A colocalization score close to 0 implies that the activities of two TCs are spatially exclusive.

## Cross-study projection of TCs on single-cell transcriptional profiles

We analyzed single-cell transcriptional data consisting of 71,965 cell-type-annotated profiles from the study on patients with HGSOC (study id GSE158722). Of these, 63,793 profiles were categorized into 21 distinct cell types. To explore the remaining unannotated profiles, we randomly selected 10% for further analysis. We first removed the first principal component from the profile correlation matrix to minimize pervasive background signals. We then conducted a cross-dataset projection of TCs onto these profiles, calculating activity scores across diverse cellular contexts.

## Univariate survival analysis using transcriptional profiles of patients with ovarian clear cell carcinoma

Expression profiles and survival data for ovarian clear cell carcinoma (OCCC) patients from Bolton et al. were obtained from the study's GitHub repository: https://github.com/kbolton-lab/Bolton_OCCC (*Wiley and Tran, 2022*; *Bolton et al., 2022*). Next, we calculated the activity scores for all TCs across all samples after performing preprocessing as described above. Subsequently, we conducted the univariate association analysis, as outlined earlier, using survival information from 120 patients

for whom transcriptional profiles were available. The results are summarized by displaying log-transformed p-values along with the signs of the coefficients.

