## [Editor Report · eLife Assessment]

This **valuable** study uses consensus-independent component analysis to highlight transcriptional components (TC) in high-grade serous ovarian cancers (HGSOC). The study presents a **convincing** preliminary finding by identifying a TC linked to synaptic signaling that is associated with shorter overall survival in HGSOC patients, highlighting the potential role of neuronal interactions in the tumour microenvironment. This finding is corroborated by comparing spatially resolved transcriptomics in a small-scale study; a weakness is it being descriptive, non-mechanistic, and requires experimental validation.

---

## [Referee Report · Reviewer #1 (Public review)]

Summary:

This manuscript explores the transcriptional landscape of high-grade serous ovarian cancer (HGSOC) using consensus-independent component analysis (c-ICA) to identify transcriptional components (TCs) associated with patient outcomes. The study analyzes 678 HGSOC transcriptomes, supplemented with 447 transcriptomes from other ovarian cancer types and noncancerous tissues. By identifying 374 TCs, the authors aim to uncover subtle transcriptional patterns that could serve as novel drug targets. Notably, a transcriptional component linked to synaptic signaling was associated with shorter overall survival (OS) in patients, suggesting a potential role for neuronal interactions in the tumor microenvironment. Given notable weaknesses like lack of validation cohort or validation using other platforms (other than the 11 samples with ST), the data is considered highly descriptive and preliminary.

The study reveals significant findings by identifying a transcriptional component (TC121) associated with synaptic signaling, which is linked to shorter survival in patients with high-grade serous ovarian cancer, highlighting the potential role of neurons in the tumor microenvironment. However, the evidence could be strengthened by experimental validation to confirm the functional roles of key genes within TC121 and further exploration of its spatial aspects, including deeper analysis of neuronal and synaptic and other neuronal gene expression.

Strengths:

Innovative Methodology:

The use of c-ICA to dissect bulk transcriptomes into independent components is a novel approach that allows for the identification of subtle transcriptional patterns that may be overshadowed in traditional analyses.

Comprehensive Data Integration:

The study integrates a large dataset from multiple public repositories, enhancing the robustness of the findings. The inclusion of spatially resolved transcriptomes adds a valuable dimension to the analysis.

Clinical Relevance:

The identification of a synaptic signaling-related TC associated with poor prognosis highlights a potential new avenue for therapeutic intervention, emphasizing the role of the tumor microenvironment in cancer progression.

Weaknesses:

Mechanistic Insights:

While the study identifies TCs associated with survival, it provides limited mechanistic insights into how these components influence cancer progression. Further experimental validation is necessary to elucidate the underlying biological processes.

Generalizability:

The findings are primarily based on transcriptomic data from HGSOC. It remains unclear how these results apply to other subtypes of ovarian cancer or different cancer types.

Innovative Methodology:

Requires more validation using different platforms (IHC) to validate the performance of this bulk derived data. Also, the lack of control on data quality is a concern.

Clinical Application:

Although the study suggests potential drug targets, the translation of these findings into clinical practice is not addressed. Probably given lack of some QA/QC procedures it'll be hard to translate these results. Future studies should focus on validating these targets in clinical settings.

---

## [Referee Report · Reviewer #2 (Public review)]

Summary:

Consensus-independent component analysis and closely related methods have previously been used to reveal components of transcriptomic data which are not captured by principal component or gene-gene coexpression analyses.

Here, the authors asked whether applying consensus-independent component analysis (c-ICA) to published high-grade serous ovarian cancer (HGSOC) microarray-based transcriptomes would reveal subtle transcriptional patterns which are not captured by existing molecular omics classifications of HGSOC.

Statistical associations of these (hitherto masked) transcriptional components with prognostic outcomes in HGSOC would lead to additional insights into underlying mechanisms and, coupled with corroborating evidence from spatial transcriptomics, are proposed for further investigation.

This approach is complementary to existing transcriptomics classifications of HGSOC.

The authors have previously applied the same approach in colorectal carcinoma (for example, Knapen et al. (2024) Commun. Med).

Strengths:

Overall, this study describes a solid data-driven description of c-ICA-derived transcriptional components that the authors identified in HGSOC microarray transcriptomics data, supported by detailed methods and supplementary documentation.

The biological interpretation of transcriptional components is convincing based on (data-driven) permutation analysis and a suite of analyses of association with copy-number, gene sets, and prognostic outcomes.

The resulting annotated transcriptional components have been made available in a searchable online format.

For the highlighted transcriptional component which has been annotated as related to synaptic signalling, the detection of the transcriptional component among 11 published spatial transcriptomics samples from ovarian cancers is compelling and supports the need for further mechanistic follow-up.

Further comments:

This revised version includes a suite of comparisons between the c-ICA-derived components and existing published transcriptomic/genomic-based classifications of ovarian cancers. Newly described components will require experimental validation, as acknowledged by the authors.

Here, the authors primarily interpret the c-ICA transcriptional components as a deconvolution of bulk transcriptomics due to the presence of cells from tumour cells and the tumour microenvironment.

In this revised version, the authors additionally investigate their TC scores in single cells from a published HGSOC single-cell RNAseq dataset, highlighting examples of TC scores within and between cell types.

c-ICA is not explicitly a deconvolution method with respect to cell types: the transcriptional components do not necessarily correspond to distinct cell types, and may reflect differential dysregulation within a cell type. This application of c-ICA for the purpose of data-driven deconvolution of cell populations is distinct from other deconvolution methods which explicitly use a prior cell signature matrix.

---

## [Author Response]

The following is the authors’ response to the original reviews

**eLife Assessment**
This valuable study uses consensus-independent component analysis to highlight transcriptional components (TC) in high-grade serous ovarian cancers (HGSOC). The study presents a convincing preliminary finding by identifying a TC linked to synaptic signaling that is associated with shorter overall survival in HGSOC patients, highlighting the potential role of neuronal interactions in the tumour microenvironment. This finding is corroborated by comparing spatially resolved transcriptomics in a small-scale study; a weakness is in being descriptive, non-mechanistic, and requiring experimental validation.”

We sincerely thank the editors for their valuable and constructive feedback. We are grateful for the recognition of our findings and the importance of identifying transcriptional components in high-grade serous ovarian cancers.

We acknowledge the editors’ observation regarding the descriptive nature of our study and its limited mechanistic depth. We agree that additional experimental validation would further strengthen our conclusions. We are planning and executing the experiments for a future study to provide mechanistic insights into the associations found in this study. In addition, recent reviews focused on the emerging field of cancer neuroscience emphasize the early stages the field is in, specifically in terms of a mechanistic understanding of the contributions of tumor-infiltrating nerves in tumor initiation and progression (Amit et al., 2024; Hwang et al., 2024). Nonetheless, we wish to emphasize that emerging mechanistic preclinical studies have demonstrated the influence of tumour-infiltrating nerves on disease progression (Allen et al., 2018; Balood et al., 2022; Darragh et al., 2024; Globig et al., 2023; Jin et al., 2022; Restaino et al., 2023; Zahalka et al., 2017). Several of these studies include contributions from our co-authors and feature in vitro and in vivo research on head and neck squamous cell carcinoma as well as high-grade serous ovarian carcinoma samples. This study further strengthens the preclinical work by showing in patient data, the potential relevance of neuronal signaling on disease outcome.

For instance, Restiano et al. (2023) demonstrated that substance P, released from tumour-infiltrating nociceptors, potentiates MAP kinase signaling in cancer cells, thereby driving disease progression. Crucially, this effect was shown to be reversible in vivo by blocking the substance P receptor (Restaino et al., 2023). These findings offer compelling evidence of the role of tumour innervation in cancer biology.

Our current study in tumor samples of patients with high-grade serous ovarian cancer identifies a transcriptional component that is enriched for genes for which the protein is located in the synapse. We believe that the previously published mechanistic insights support our findings and suggest that this transcriptional component could serve as a valuable screening tool to identify innervated tumours based on bulk transcriptomes. Clinically, this information is highly relevant, as patients with innervated tumours may benefit from alternate therapeutic strategies targeting these innervations.

**Reviewer #1 (Public review)**
This manuscript explores the transcriptional landscape of high-grade serous ovarian cancer (HGSOC) using consensus-independent component analysis (c-ICA) to identify transcriptional components (TCs) associated with patient outcomes. The study analyzes 678 HGSOC transcriptomes, supplemented with 447 transcriptomes from other ovarian cancer types and noncancerous tissues. By identifying 374 TCs, the authors aim to uncover subtle transcriptional patterns that could serve as novel drug targets. Notably, a transcriptional component linked to synaptic signaling was associated with shorter overall survival (OS) in patients, suggesting a potential role for neuronal interactions in the tumour microenvironment. Given notable weaknesses like lack of validation cohort or validation using another platform (other than the 11 samples with ST), the data is considered highly descriptive and preliminary.Strengths:(1) Innovative Methodology:The use of c-ICA to dissect bulk transcriptomes into independent components is a novel approach that allows for the identification of subtle transcriptional patterns that may be overshadowed in traditional analyses.

We thank the reviewer for recognizing the strengths and novelty of our study. We appreciate the positive feedback on using consensus-independent component analysis (c-ICA) to decompose bulk transcriptomes, which allowed us to detect subtle transcriptional signals often overlooked in traditional analyses.

(2) Comprehensive Data Integration:The study integrates a large dataset from multiple public repositories, enhancing the robustness of the findings. The inclusion of spatially resolved transcriptomes adds a valuable dimension to the analysis.

We thank the reviewer for recognizing the robustness of our study through comprehensive data integration. We appreciate the acknowledgment of our efforts to leverage a large, multi-source dataset, as well as the additional insights gained from spatially resolved transcriptomes. We consider this integrative approach enhances the depth of our analysis and contributes to a more nuanced understanding of the tumour microenvironment.

(3) Clinical Relevance:The identification of a synaptic signaling-related TC associated with poor prognosis highlights a potential new avenue for therapeutic intervention, emphasizing the role of the tumour microenvironment in cancer progression.

We appreciate the recognition of the clinical implications of our findings. The identification of a synaptic signaling-related transcriptional component associated with poor prognosis underscores the potential for novel therapeutic targets within the tumour microenvironment. We agree that this insight could open new avenues for intervention and further highlights the role of neuronal interactions in cancer progression.

Weaknesses:(1) Mechanistic Insights:While the study identifies TCs associated with survival, it provides limited mechanistic insights into how these components influence cancer progression. Further experimental validation is necessary to elucidate the underlying biological processes.

We acknowledge the point regarding the limited mechanistic insights provided in our study. We agree that further experimental validation would significantly enhance our understanding of how the biological processes captured by these transcriptional components influence cancer progression. We are planning and executing the experiments for a future study to provide mechanistic insights into the associations found in this study.

Our analyses were performed on publicly available bulk and spatial resolved expression profiles. To investigate the mechanistic insights in future studies, we plan to integrate spatial transcriptomic data with immunohistochemical analysis of the same tumour samples to validate our findings. Additionally, we have initiated efforts to set up in vitro co-cultures of neurons and ovarian cancer cells. These co-cultures will enable us to investigate how synaptic signaling impacts ovarian cancer cell behavior.

(2) Generalizability:The findings are primarily based on transcriptomic data from HGSOC. It remains unclear how these results apply to other subtypes of ovarian cancer or different cancer types.

To respond to this remark, we utilized survival data from Bolton et al. (2022) and TCGA to investigate associations between TC activity scores and overall survival of patients with ovarian clear cell carcinoma, the second most common subtype of epithelial ovarian cancer, and other cancer types respectively. However, we acknowledge the limitations of TCGA survival data, as highlighted in the referenced article (https://www.ncbi.nlm.nih.gov/pmc/articles/PMC8726696/). Additionally, as shown in Figure 5, we provided evidence of TC121 activity across various cancer types, suggesting broader relevance. For the results of the analyses mentioned above, please refer to our response to remark 1.3 of the recommendation section (page 4).

(3) Innovative Methodology:Requires more validation using different platforms (IHC) to validate the performance of this bulk-derived data. Also, the lack of control over data quality is a concern.

We acknowledge the value of validating our results with alternative platforms such as IHC. We are planning and executing the experiments for a future study to provide mechanistic insights into the associations found in this study.

We implemented regarding data quality control, the following measures to ensure the reliability of our analysis:

Bulk Transcriptional Profiles: To assess data quality, we conducted principal component analysis (PCA) on the sample Pearson product-moment correlation matrix. The first principal component (PCqc), which explains approximately 80-90% of the variance, was used to distinguish technical variability from biological signals (Bhattacharya et al., 2020). Samples with a correlation coefficient below 0.8 relative to PCqc were identified as outliers and excluded. Additionally, MD5 hash values were generated for each CEL file to identify and remove duplicate samples. Expression values were standardized to a mean of zero and a variance of one for each gene to minimize probeset- or gene-specific variability across datasets (GEO, CCLE, GDSC, and TCGA).

Spatial Transcriptional Profiles: PCA was also applied to spatial transcriptomic data for quality control. Only samples with consistent loading factor signs for the first principal component across all individual spot profiles were retained. Samples failing this criterion were excluded from further analyses.

(4) Clinical Application:Although the study suggests potential drug targets, the translation of these findings into clinical practice is not addressed. Probably given the lack of some QA/QC procedures it'll be hard to translate these results. Future studies should focus on validating these targets in clinical settings.”

Regarding clinical applications, we acknowledge the importance of further exploring strategies targeting synaptic signaling and neurotransmitter release in the tumour microenvironment (TME). As partially discussed in the first version of the manuscript, drugs such as ifenprodil and lamotrigine—commonly used to treat neuronal disorders—can block glutamate release, thereby inhibiting subsequent synaptic signaling. Additionally, the vesicular monoamine transporter (VMAT) inhibitor reserpine blocks the formation of synaptic vesicles (Reid et al., 2013; Williams et al., 2001). Previous in vitro studies with HGSOC cell lines demonstrated that ifenprodil significantly reduced cancer cell proliferation, while reserpine triggered apoptosis in cancer cells (North et al., 2015; Ramamoorthy et al., 2019). The findings highlight the potential of such approaches to disrupt synaptic neurotransmission in the TME.

To address potential translation of our findings into clinical practice more comprehensively, we have included additional details in the manuscript:

Section discussion, page 16, lines 338-341:

“This interaction can be targeted with pan-TRK inhibitors such as entrectinib and larotrectinib. Both drugs are showing promising results in multiple phase II trials, including ovarian cancer and breast cancer patients. Furthermore, a TRKB-specific inhibitor was developed (ANA-12), but has not been subjected to any clinical trials in cancer so far (Ardini et al., 2016; Burris et al., 2015; Drilon et al., 2018, 2017).”

On page 17, lines 361-374:

“Strategies to disrupt neuronal signaling and neurotransmitter release in neurons target key elements of excitatory neurotransmission, such as calcium flux and vesicle formation. Drugs like ifenprodil and lamotrigine, commonly used to treat neuronal disorders, block glutamate release and subsequent neuronal signaling. Additionally, the vesicular monoamine transporter (VMAT) inhibitor reserpine prevents synaptic vesicle formation (Reid et al., 2013; Williams, 2001). In vitro studies with HGSOC cell lines have demonstrated that ifenprodil significantly inhibits tumour proliferation, while reserpine induces apoptosis in cancer cells (North et al., 2015; Ramamoorthy et al., 2019). These approaches hold promise for inhibiting neuronal signaling and interactions in the TME.”

**Reviewer #2 (Public review):**
Summary:Consensus-independent component analysis and closely related methods have previously been used to reveal components of transcriptomic data that are not captured by principal component or gene-gene coexpression analyses.Here, the authors asked whether applying consensus-independent component analysis (c-ICA) to published high-grade serous ovarian cancer (HGSOC) microarray-based transcriptomes would reveal subtle transcriptional patterns that are not captured by existing molecular omics classifications of HGSOC.Statistical associations of these (hitherto masked) transcriptional components with prognostic outcomes in HGSOC could lead to additional insights into underlying mechanisms and, coupled with corroborating evidence from spatial transcriptomics, are proposed for further investigation.This approach is complementary to existing transcriptomics classifications of HGSOC.The authors have previously applied the same approach in colorectal carcinoma (Knapen et al. (2024) Commun. Med).Strengths:(1) Overall, this study describes a solid data-driven description of c-ICA-derived transcriptional components that the authors identified in HGSOC microarray transcriptomics data, supported by detailed methods and supplementary documentation.

We thank the reviewer for acknowledging the strength of our data-driven approach and the use of consensus-independent component analysis (c-ICA) to identify transcriptional components within HGSOC microarray data. We aimed to provide comprehensive methodological detail and supplementary documentation to support the reproducibility and robustness of our findings. We believe this approach allows for the identification of subtle transcriptional signals that might have been overlooked by traditional analysis methods.

(2) The biological interpretation of transcriptional components is convincing based on (data-driven) permutation analysis and a suite of analyses of association with copy-number, gene sets, and prognostic outcomes.

We appreciate the positive feedback on the biological interpretation of our transcriptional components. We are pleased that our approach, which includes data-driven permutation testing and analyses of associations with copy-number alterations, gene sets, and prognostic outcomes, was found to be convincing. These analyses were integral to enhancing our findings’ robustness and biological relevance.

(3) The resulting annotated transcriptional components have been made available in a searchable online format.

Thank you for this important positive remark.

(4) For the highlighted transcriptional component which has been annotated as related to synaptic signalling, the detection of the transcriptional component among 11 published spatial transcriptomics samples from ovarian cancers appears to support this preliminary finding and requires further mechanistic follow-up.

Thank you for acknowledging the accessibility of our annotated transcriptional components. We prioritized making these data available in a searchable online format to facilitate further research and enable the community to explore and validate our findings.

Weaknesses:(1) This study has not explicitly compared the c-ICA transcriptional components to the existing reported transcriptional landscape and classifications for ovarian cancers (e.g. Smith et al Nat Comms 2023; TCGA Nature 2011; Engqvist et al Sci Rep 2020) which would enable a further assessment of the additional contribution of c-ICA - whether the cICA approach captured entirely complementary components, or whether some components are correlated with the existing reported ovarian transcriptomic classifications.

We acknowledge the reviewer’s insightful suggestion to compare our c-ICA-derived transcriptional components with previously reported ovarian cancer classifications, such as those from Smith et al. (2023), TCGA (2011), and Engqvist et al. (2020). To address this, we incorporated analyses comparing the activity scores of our transcriptional components with these published landscapes and classifications, particularly focusing on any associations with overall survival. Additionally, we evaluated correlations between gene signatures from a subset of these studies and our identified TCs, enhancing our understanding of the unique contributions of the c-ICA approach. Please refer to our response to remark 10 for the results of these analyses.

(2) Here, the authors primarily interpret the c-ICA transcriptional components as a deconvolution of bulk transcriptomics due to the presence of cells from tumour cells and the tumour microenvironment.However, c-ICA is not explicitly a deconvolution method with respect to cell types: the transcriptional components do not necessarily correspond to distinct cell types, and may reflect differential dysregulation within a cell type. This application of c-ICA for the purpose of data-driven deconvolution of cell populations is distinct from other deconvolution methods that explicitly use a prior cell signature matrix.”

We acknowledge that c-ICA, unlike traditional deconvolution methods, is not specifically designed for cell-type deconvolution and does not rely on a predefined cell signature matrix. While we explored the transcriptional components in the context of tumour and microenvironmental interactions, we agree that these components may not correspond directly to distinct cell types but rather reflect complex patterns of dysregulation, potentially within individual cell populations.

Our goal with c-ICA was to uncover hidden transcriptional patterns possibly influenced by cellular heterogeneity. However, we recognize these patterns may also arise from regulatory processes within a single cell type. To investigate further, we used single-cell transcriptional data (~60,000 cell-types annotated profiles from GSE158722) and projected our transcriptional components onto these profiles to obtain activity scores, allowing us to assess each TC’s behavior across diverse cellular contexts after removing the first principal component to minimize background effects. Please refer to our response to remark 2.2 in the recommendations to the authors (page 14) for the results of this analysis.

References

Allen JK, Armaiz-Pena GN, Nagaraja AS, Sadaoui NC, Ortiz T, Dood R, Ozcan M, Herder DM, Haemerrle M, Gharpure KM, Rupaimoole R, Previs R, Wu SY, Pradeep S, Xu X, Han HD, Zand B, Dalton HJ, Taylor M, Hu W, Bottsford-Miller J, Moreno-Smith M, Kang Y, Mangala LS, Rodriguez-Aguayo C, Sehgal V, Spaeth EL, Ram PT, Wong ST, Marini FC, Lopez-Berestein G, Cole SW, Lutgendorf SK, diBiasi M, Sood AK. 2018. Sustained adrenergic signaling promotes intratumoral innervation through BDNF induction. Cancer Res 78 (12):3233-3242.

Ardini E, Menichincheri M, Banfi P, Bosotti R, Ponti CD, Pulci R, Ballinari D, Ciomei M, Texido G, Degrassi A, Avanzi N, Amboldi N, Saccardo MB, Casero D, Orsini P, Bandiera T, Mologni L, Anderson D, Wei G, Harris J, Vernier J-M, Li G, Felder E, Donati D, Isacchi A, Pesenti E, Magnaghi P, Galvani A. 2016. Entrectinib, a Pan–TRK, ROS1, and ALK Inhibitor with activity in multiple molecularly defined cancer Indications. Mol Cancer Ther 15:628–639.

Balood M, Ahmadi M, Eichwald T, Ahmadi A, Majdoubi A, Roversi Karine, Roversi Katiane, Lucido CT, Restaino AC, Huang S, Ji L, Huang K-C, Semerena E, Thomas SC, Trevino AE, Merrison H, Parrin A, Doyle B, Vermeer DW, Spanos WC, Williamson CS, Seehus CR, Foster SL, Dai H, Shu CJ, Rangachari M, Thibodeau J, Rincon SVD, Drapkin R, Rafei M, Ghasemlou N, Vermeer PD, Woolf CJ, Talbot S. 2022. Nociceptor neurons affect cancer immunosurveillance. Nature 611:405–412.

Bhattacharya A, Bense RD, Urzúa-Traslaviña CG, Vries EGE de, Vugt MATM van, Fehrmann RSN. 2020. Transcriptional effects of copy number alterations in a large set of human cancers. Nat Commun 11:715.

Burris HA, Shaw AT, Bauer TM, Farago AF, Doebele RC, Smith S, Nanda N, Cruickshank S, Low JA, Brose MS. 2015. Abstract 4529: Pharmacokinetics (PK) of LOXO-101 during the first-in-human Phase I study in patients with advanced solid tumors: Interim update. Cancer Res 75:4529–4529.